# Inference from Real-World Sparse Measurements

**Arnaud Pannatier**                                              *arnaud.pannatier@idiap.ch*
*Idiap Research Institute, Martigny, Switzerland*
*École Polytechnique Fédérale de Lausanne, Lausanne, Switzerland*

**Kyle Matoba**
*Idiap Research Institute, Martigny, Switzerland*
*École Polytechnique Fédérale de Lausanne, Lausanne, Switzerland*

**François Fleuret**
*Université de Genève, Geneva, Switzerland*

**Reviewed on OpenReview:** *https://openreview.net/forum?id=y9IDfODRns*

## Abstract

Real-world problems often involve complex and unstructured sets of measurements, which occur when sensors are sparsely placed in either space or time. Being able to model this irregular spatiotemporal data and extract meaningful forecasts is crucial. Deep learning architectures capable of processing sets of measurements with positions varying from set to set, and extracting readouts anywhere are methodologically difficult. Current state-of-the-art models are graph neural networks and require domain-specific knowledge for proper setup.

We propose an attention-based model focused on robustness and practical applicability, with two key design contributions. First, we adopt a ViT-like transformer that takes both context points and read-out positions as inputs, eliminating the need for an encoder-decoder structure. Second, we use a unified method for encoding both context and read-out positions. This approach is intentionally straightforward and integrates well with other systems. Compared to existing approaches, our model is simpler, requires less specialized knowledge, and does not suffer from a problematic bottleneck effect, all of which contribute to superior performance.

We conduct in-depth ablation studies that characterize this problematic bottleneck in the latent representations of alternative models that inhibit information utilization and impede training efficiency. We also perform experiments across various problem domains, including high-altitude wind nowcasting, two-day weather forecasting, fluid dynamics, and heat diffusion. Our attention-based model consistently outperforms state-of-the-art models in handling irregularly sampled data. Notably, our model reduces the root mean square error (RMSE) for wind nowcasting from 9.24 to 7.98 and for heat diffusion tasks from 0.126 to 0.084.

## 1 Introduction

Deep learning (DL) has emerged as a powerful tool for modeling dynamical systems by leveraging vast amounts of data available in ways that traditional solvers cannot. This has led to a growing reliance on DL models in weather forecasting, with state-of-the-art results in precipitation nowcasting (Suman et al., 2021; Shi et al., 2017) and performance on par with traditional partial differential equation (PDE) solvers in medium-term forecasting (Lam et al., 2022).

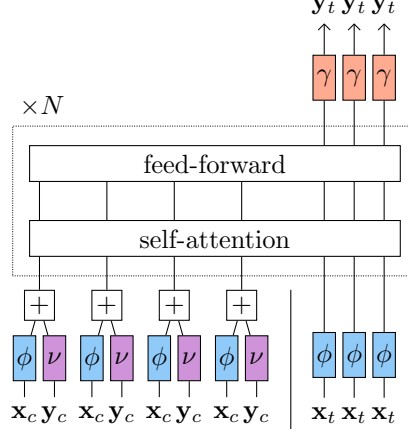

Figure 1: Multi-layer Self-Attention

| Architecture | Performance | Simplicity | Domain Knowledge Agnostic |
|---|---|---|---|
| CNP | ✗ | ✓✓ | ✓✓ |
| GEN | ✓ | ✗ | ✗ |
| TFS | ✓ | ✓ | ✓✓ |
| MSA (Ours) | ✓✓ | ✓✓ | ✓✓ |

Table 1: Comparison between our approach and the different baselines. Multi-Layer Self-Attention (MSA) achieves good performance while being simple to implement and does not require practitioner knowledge for proper setup.

However, these applications are currently limited to data represented as images or on regular grids, where models such as convolutional networks or graph neural networks are used. In contrast, various real-world data often come from irregularly placed or moving sensors, which means custom architectures are needed to handle it effectively.

An example that can benefit significantly from such an architecture is Air Traffic Control (ATC). ATC needs reliable weather forecasts to manage airspace efficiently. This is particularly true for wind conditions, as planes are highly sensitive to wind and deviations from the initial flight plan can be costly and pose safety hazards. DL models are a promising candidate for producing reliable wind forecasts as a large amount of data is collected from airplanes that broadcast wind speed measurements with a four-second frequency. A model that can effectively model wind speeds using data collected from airplanes, should be able to decode anywhere in space, as we aim to predict wind conditions at future locations of the airplane, conditioned on past measurements taken by that specific airplane or neighboring ones in a permutation invariant manner.

To meet these requirements, we introduce a Multi-layer Self-Attention model (MSA) and compare it to different baselines [Table 1]: Conditional Neural Processes (CNP) (Garnelo et al., 2018), Graph Element Networks (GEN) (Alet et al., 2019) and a transformer encoder-decoder baseline (TFS) that we developed. While all of these models possess the aforementioned characteristics, they each adopt distinct strategies for representing measurements within the latent space. CNP models use a single vector as a summary of encoded measures, while GEN models map the context to a graph, based on their distance to its nodes. MSA keeps one latent vector per encoded measurement and can access them directly for forecasting. This latent representation is better, as it does not create a bottleneck. We show that due to that architectural choice, both baselines can fail, in certain cases, to use information from the context they condition on.

Our approach offers better performance than its competitors and is conceptually simpler as it does not require an encoder-decoder structure. To evaluate the effectiveness of our approach, we conducted experiments on high-altitude wind nowcasting, heat diffusion, fluid dynamics, and two-day weather forecasting. Several additional ablation studies show the impact of different architectural choices.

The main contributions of this work are summarized below:

- We develop an attention-based model that can generate prediction anywhere in the space conditioned on a set of measurements.

- We propose a novel encoding scheme using a shared MLP encoder to map context and target positions, improving forecasting performance and enhancing the model's understanding of spatial patterns.

- We evaluate our method on a set of challenging tasks with data irregularly sampled in space: high-altitude wind nowcasting, two-day weather forecasting, heat diffusion, and fluid dynamics.

- We examine the differences between models, and explain the impact of design choices such as latent representation bottlenecks on the final performance of the trained models.

## 2 Related Works

DL performance for weather forecasting has improved in recent years, with DL models increasingly matching or surpassing the performance of traditional PDE-based systems. Initially applied to precipitation nowcasting based on 2D radar images (Suman et al., 2021; Shi et al., 2017), DL-based models have recently surpassed traditional methods for longer forecast periods (Lam et al., 2022). These methods usually work with highly structured data. Radar precipitation data, for example, can be organized as images and analyzed using convolutional neural networks. For 3D regular spherical grid data, graph neural networks or spherical CNNs are employed (Lam et al., 2022; Esteves et al., 2023). However, in our study, the data set is distributed sparsely in space, which hinders the use of these traditional architectures. The use of DL for modeling dynamical systems, in general, has also seen recent advancements (Li et al., 2021; Gupta & Brandstetter, 2022; Pfaff et al., 2020) but most approaches in this field typically operate on regularly-spaced data or irregular but fixed mesh.

Neural Processes (Garnelo et al., 2018; Kim et al., 2019), Graph Element Networks (Alet et al., 2019) and attention-based models (Vaswani et al., 2017) are three DL-based approaches that are capable of modeling sets of data changing from set to set. In this study, we conduct a comparison of these models by selecting a representative architecture from each category. Additionally, attention-based models have been previously adapted for set classification tasks (Lee et al., 2019), and here we adapt them to generate forecasts.

Pannatier et al. (2021) use a kernel-based method for wind nowcasting based on flight data. This method incorporates a distance metric with learned parameters to combine contexts for prediction at any spatial location. However, a notable limitation of this technique is that its forecasts are constrained to the convex hull of the input measurements, preventing accurate extrapolation. We evaluate the efficacy of our method compared to this approach, along with the distinct outcomes obtained, in Appendix F of the supplementary material.

While previous studies have utilized transformers for modeling physical systems (Geneva & Zabaras, 2022), time series (Li et al., 2019) or trajectory prediction (Girgis et al., 2022; Nayakanti et al., 2023; Yuan et al., 2021) these applications do not fully capture the specific structure of our particular domain, which involves relating two spatial processes at arbitrary points on a shared domain. Although we model temporal relationships, our approach lacks specialized treatment of time. Therefore, it does not support inherently time-based concepts like heteroskedasticity, time-series imputation, recurrence, or seasonality. Further details distinguishing our approach from other transformer-based applications are elaborated in Appendix C of the supplementary material.

## 3 Methodology

### 3.1 Context and Targets

The problem addressed in this paper is the prediction of target values given a context and a prediction target position. Data is in the form of pairs of vectors $(\mathbf{x}_c, \mathbf{y}_c)$ and $(\mathbf{x}_t, \mathbf{y}_t)$ where $\mathbf{x}_c$ and $\mathbf{x}_t$ are the position and $\mathbf{y}_c$ and $\mathbf{y}_t$ are the measurements (or values), where we use $c$ for context, $t$ for target, $x$ for spatial position and $y$ for the corresponding vector value. The positions lie in the same underlying space $\mathbf{x}_c, \mathbf{x}_t \in \mathbb{X} \subseteq \mathbb{R}^X$, but the context and target values do not necessarily. We define the corresponding spaces as $\mathbf{y}_c \in \mathbb{I} \subseteq \mathbb{R}^I$ and $\mathbf{y}_t \in \mathbb{O} \subseteq \mathbb{R}^O$, respectively, where $X, I, O$ are integers that need not be equal. The dataset comprises multiple pairs of context and target sets, each possibly varying in length. We denote the length of the $j$-th context and target set as $N_c^j$ and $N_t^j$. All models take as input a set of context pairs $\{(\mathbf{x}_c, \mathbf{y}_c)_i^j\}_{i=1}^{N_c^j}$, as well as target positions, denoted $\{(\mathbf{x}_t)_i^j\}_{i=1}^{N_t^j}$.

As an example, to transform a data set of wind speed measurements into context and target pair, we partitioned the data set into one-minute time segments and generated context and target sets with an intervening delay, as depicted in Figure 2. The underlying space, denoted by $\mathbb{X}$, corresponds to 3D Euclidean space, with both $\mathbb{I}$ and $\mathbb{O}$ representing wind speed measurements in the $x, y$ plane. The models are given a set of context points at positions $\mathbf{x}_c$ of value $\mathbf{y}_c$ and query positions $\mathbf{x}_t$ and are tasked to produce a corresponding

value $\mathbf{y}_t$ conditioned on the context. Detailed descriptions of the data set, including illustrations of the different problems, and the respective spaces for other scenarios and the ablation study can be found in Table 7 within the supplementary material.

## 3.2 Encoding Scheme

We propose in this section a novel encoding scheme for irregularly sampled data. Our approach leverages the fact that both the context measurements and target positions reside within a shared underlying space. To exploit this shared structure, we adopt a unified two-layer MLP encoder $\phi$ for mapping both the context and target position to a latent space representation. Then, we use a second MLP $\nu$ to encode the context values and add them to the encoded positions when available. This differs from the approach proposed in Garnelo et al. (2018); Alet et al. (2019) where both the context position and value are concatenated and given to an encoder, and the target position is encoded by another. The schemes are contrasted as:

$$\mathbf{e}_c = \varphi(\mathbf{x}_c, \mathbf{y}_c) \qquad (1) \qquad\qquad \mathbf{e}_c = \phi(\mathbf{x}_c) + \nu(\mathbf{y}_c) \qquad (3)$$

$$\mathbf{e}_t = \psi(\mathbf{x}_t) \qquad (2) \qquad\qquad \mathbf{e}_t = \phi(\mathbf{x}_t) \qquad (4)$$

Traditional methods          Proposed scheme

Where $\phi, \nu, \varphi, \psi$ are two hidden-layers MLPs, and $\mathbf{e}_c, \mathbf{e}_t \in \mathbb{R}^E$ are context positions and values which are concatenated and encoded, and target position which is encoded respectively.

## 3.3 Multi-layer Self-Attention (MSA, Ours)

Our proposed model, Multi-layer Self-Attention (MSA) harnesses the advantages of attention-based models [Figure 1]. MSA maintains a single latent representation per input measurement and target position, which conveys the ability to propagate gradients easily and correct errors in training quickly. MSA can access and combine target position and context measurements at the same time, which forms a flexible latent representation. Our model is similar to a transformer-encoder, as the backbone of a ViT (Dosovitskiy et al., 2020), it can be written as:

$$\{(\mathbf{l}_c)_i^j\}_{i=1}^{N_c^j}, \{(\mathbf{l}_t)_i^j\}_{i=1}^{N_t^j} = \text{Transformer-Encoder}(\{(\mathbf{e}_c)_i^j\}_{i=1}^{N_c^j}, \{(\mathbf{e}_t)_i^j\}_{i=1}^{N_t^j}) \qquad (5)$$

$$\{(\hat{\mathbf{y}}_t)_i^j\}_{i=1}^{N_t^j} = \{\gamma((\mathbf{l}_t)_i^j)\}_{i=1}^{N_t^j} \qquad (6)$$

where $\gamma$ is a two hidden-layers MLP that is used to generate readouts from the latent space. As a transformer-encoder outputs the same number of outputs as inputs, we define its output as $\{(\mathbf{l}_c)_i^j\}_{i=1}^{N_c^j}, \{(\mathbf{l}_t)_i^j\}_{i=1}^{N_t^j}$ corresponding to the latent representations of the context and target positions respectively. However, only $\{(\mathbf{l}_t)_i^j\}_{i=1}^{N_t^j}$ is used to generate the readouts.

MSA does not use positional encoding for encoding the order of the inputs. This model is permutation equivariant due to the self-attention mechanism and it uses full attention, allowing each target feature to attend to all other targets and context measurements. MSA generates all the output in one pass in a non-autoregressive way and the outputs of the model are only the units that correspond to the target positions, which are then used to compute the loss.

## 3.4 Baselines

**Transformer(s) (TFS)** We also modify an encoder-decoder transformer (TFS) model (Vaswani et al., 2017) to the task at hand. The motivation behind this was the intuitive appeal of the encoder-decoder stack for this specific problem. TFS in our approach deviates from the standard transformer in a few ways: Firstly, it does not employ causal masking in the decoder and secondly, the model forgoes the use of positional encoding for the sequence positions. It can be written as:

$$\{(\mathbf{l}_c)_i^j\}_{i=1}^{N_c^j} = \text{Transformer-Encoder}(\{(\mathbf{e}_c)_i^j\}_{i=1}^{N_c^j}) \tag{7}$$

$$\{(\mathbf{l}_t)_i^j\}_{i=1}^{N_t^j} = \text{Transformer-Decoder}(\{(\mathbf{l}_c)_i^j\}_{i=1}^{N_c^j}, \{(\mathbf{e}_t)_i^j\}_{i=1}^{N_t^j}) \tag{8}$$

$$\{(\hat{\mathbf{y}}_t)_i^j\}_{i=1}^{N_t^j} = \{\gamma((\mathbf{l}_t)_i^j)\}_{i=1}^{N_t^j} \tag{9}$$

In comparison to MSA, TFS uses an encoder-decoder architecture, which adds a layer of complexity. Moreover, it necessitates the propagation of error through two pathways, specifically through a cross-attention mechanism that lacks a residual connection to the encoder inputs.

**Graph Element Network(s) (GEN)**  Graph Element Networks (GEN) (Alet et al., 2019) represent an architecture that leverages a graph $\mathcal{G}$ as a latent representation. This graph comprises $N$ nodes situated at positions $\mathbf{x}_n$ and connected by $E$ edges. Selecting the nodes' positions and the graph's edges is critical, as these are additional parameters that require careful consideration. The node positions can be made learnable, allowing for their optimization during training through gradient descent. It's important to note that the learning rate for these learnable parameters should be meticulously chosen to ensure convergence, typically being smaller in magnitude compared to other parameters.

In the original work, edges are established using Delaunay triangulation and may be altered during training in response to shifts in node positions. The encoder function transforms measurements into a latent space. These are then aggregated to formulate the initial node values, influenced by their proximity to the nodes. Specifically, a measurement at position $\mathbf{x}_c$ impacts the initial value of a node at position $\mathbf{e}_i$ (with $i \in 1, \ldots, N$) based on a weighting function $r(\mathbf{x}_c, \mathbf{e}_i)$, which assigns greater significance to context points nearer to the graph's nodes. In the original work, this weighting function is defined as $r(\mathbf{x}_c, \mathbf{e}_i) = \text{softmax}(-\beta\|\mathbf{x}_c - \mathbf{e}_i\|)$, where $\beta$ is a learnable parameter. The initial node values undergo processing through $L$ iterations of message passing. For model readouts, the same weighting function is employed, ensuring each node is weighted according to its distance from the query point. Overall, the model can be described as follows:

$$\mathbf{e}_n = \sum_{(\mathbf{x}_c, \mathbf{e}_c) \in \mathcal{E}} r(\mathbf{x}_c, \mathbf{x}_n)\mathbf{e}_c \qquad \forall n \in \{1, \ldots, N\} \tag{10}$$

$$\{(\mathbf{l}_n)_i\}_{i=1}^N = \text{Message-Passing}(\{(\mathbf{e}_n)_i\}_{i=1}^N, \mathcal{G}, L) \tag{11}$$

$$(\mathbf{l}_t)_i^j = \sum_{(\mathbf{x}_n, \mathbf{l}_n) \in \mathcal{L}} r(\mathbf{x}_n, \mathbf{x}_t)\mathbf{l}_n \qquad \forall i \in \{1, \ldots, N_t^j\} \tag{12}$$

$$\{(\hat{\mathbf{y}}_t)_i^j\}_{i=1}^{N_t^j} = \{\gamma((\mathbf{l}_t)_i^j)\}_{i=1}^{N_t^j} \tag{13}$$

Where $\mathcal{E} = \{(\mathbf{x}_c, \mathbf{e}_c)_i^j\}_{i=1}^{N_c^j}$ is the set of encoded context and their position and $\mathcal{L} = \{(\mathbf{x}_n, \mathbf{l}_n)_i\}_{i=1}^N$ is the set of the graph nodes and their position.

GEN's inductive bias is that a single latent vector summarizes a small part of the space. As it includes a distance-based encoding and decoding scheme, the only way for the model to learn non-local patterns is through message passing. This model was originally designed with a simple message-passing scheme. But it can easily be extended to a broad family of graph networks by using different message-passing schemes, including ones with attention. We present some related experiments in Appendix K of the supplementary material.

**Conditional Neural Process(es) (CNP)**  CNP (Garnelo et al., 2018) encodes the whole context as a single latent vector. They can be seen as a subset of GEN. Specifically, a CNP is a GEN with a graph with a single node and no message passing. While CNP possesses the desirable property of being able to model any permutation-invariant function (Zaheer et al., 2017), their expressive capability is constrained by the single node architecture (Kim et al., 2019). Despite this, CNP serves as a valuable baseline and is considerably less computationally intensive.

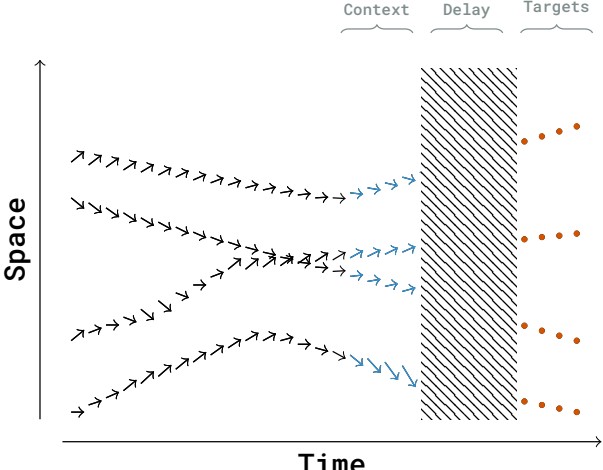

Figure 2: Description of the context and target sets in the wind nowcasting case. The context set and the target set are time slices separated by a delay, which corresponds to the forecasting window. The underlying space is in that case $\mathbb{X} \subseteq \mathbb{R}^3$ and the context values and target values both represent wind speed and belong to the same space $\mathbb{I} = \mathbb{O} \subseteq \mathbb{R}^2$.

Figure 3: Decay of precision in the wind nowcasting case. RMSE of the different models depending on the forecast duration (lower is better). We ran three experiments varying the pseudorandom number generator seeds for each time window and each model to measure the standard deviation. The error does not increase drastically over the first two hours because of the persistence of the wind and the context values are good predictors of the targets in that regime.

$$\mathbf{l}_c = \mathrm{mean}(\{(\mathbf{e}_c)_i^j\}_{i=1}^{N_c^j}) \tag{14}$$

$$\{(\mathbf{l}_t)_i^j\}_{i=1}^{N_t^j} = \{\eta\left(\mathbf{l}_t, (\mathbf{e}_t)_i^j\right)\}_{i=1}^{N_t^j} \tag{15}$$

$$\{(\hat{\mathbf{y}}_t)_i^j\}_{i=1}^{N_t^j} = \{\gamma((\mathbf{l}_t)_i^j)\}_{i=1}^{N_t^j} \tag{16}$$

Where $\eta$ is a simple linear projection.

## 4 Experiments

Our experiments aim to benchmark the performance of our models on various data sets with irregularly sampled data. The first task focuses on high-altitude wind nowcasting. The second task is on heat diffusion. Additionally, we evaluate our models on fluid flows, considering both a steady-state case governed by the Darcy Flow equation and a dynamic case modeling the Navier-Stokes equation in an irregularly spaced setting. Finally, we compare the models on a weather forecasting task, utilizing irregularly sampled measurements from the ERA5 data set (Hersbach et al., 2023) to predict wind conditions two days ahead.

For the Wind Nowcasting Experiment, the data set, described in Section 3.1, consists of wind speed measurements collected by airplanes with a sampling frequency of four seconds. We evaluate our models on this data set [Table 2] and we assess the models' performance as a function of forecast duration, as depicted in Figure 3, and against different metrics [Table 3] to ensure the robustness of our approach. We select model configurations with approximately 100,000 parameters and run each model using three different random seeds in both cases. Our results indicate that attention-based models consistently outperform other models for most forecast durations, except for in the 6-hour range. Notably, we found that the Gaussian Kernel Averaging (GKA) model used in previous work (Pannatier et al., 2021) achieves satisfactory performance, despite its theoretical limitations, which we analyze in Appendix F of the supplementary material. Moreover, our findings suggest that attention-based models, particularly MSA and TFS, exhibit superior performance

Table 2: Validation RMSE of the High-Altitude Wind Nowcasting, Poisson, Navier-Stokes, and Darcy Flow equation and the weather forecasting task. Each model ran for 10, 2000, 1000, 100, and 100 epochs respectively on an NVIDIA GeForce GTX 1080 Ti. The low number of epochs for wind nowcasting is due to the amount of data which is considerably larger than in the other experiments. The standard deviation is computed over 3 runs. We present here the original implementation of CNP and GEN compared with TFS and MSA with sharing weights for the position. More details can be found in Table 2 of the supplementary material. We choose the configuration of the models so that every model has a comparable number of parameters. We underline the best models for each size and indicate in bold the best model overall.

| Architecture | # of Params. | Wind Nowcasting | Poisson Equation | Navier-Stokes Equation | Darcy Flow Equation | ERA5 |
|---|---|---|---|---|---|---|
| **CNP** | 5k | $11.94_{\pm\ 0.78}$ | $0.33_{\pm\ 0.004}$ | $0.701_{\pm\ 0.0023}$ | $0.0311_{\pm\ 0.0008}$ | $2.129_{\pm\ 0.0039}$ |
|  | 20k | $10.19_{\pm\ 1.83}$ | $0.32_{\pm\ 0.003}$ | $0.672_{\pm\ 0.0011}$ | $0.0295_{\pm\ 0.0002}$ | $2.117_{\pm\ 0.0018}$ |
|  | 100k | $10.17_{\pm\ 1.24}$ | $0.33_{\pm\ 0.003}$ | $0.656_{\pm\ 0.0007}$ | $0.0286_{\pm\ 0.0001}$ | $2.110_{\pm\ 0.0002}$ |
| **GEN** | 5k | $11.02_{\pm\ 3.19}$ | $0.12_{\pm\ 0.006}$ | $0.604_{\pm\ 0.0010}$ | $0.0304_{\pm\ 0.0003}$ | $2.132_{\pm\ 0.0035}$ |
|  | 20k | $9.98_{\pm\ 0.76}$ | $0.13_{\pm\ 0.014}$ | $0.599_{\pm\ 0.0006}$ | $0.0296_{\pm\ 0.0002}$ | $2.124_{\pm\ 0.0031}$ |
|  | 100k | $9.56_{\pm\ 0.21}$ | $0.16_{\pm\ 0.049}$ | $0.596_{\pm\ 0.0005}$ | $0.0294_{\pm\ 0.0001}$ | $2.121_{\pm\ 0.0005}$ |
| **TFS (Ours, baseline)** | 5k | $8.30_{\pm\ 0.03}$ | $0.15_{\pm\ 0.036}$ | $0.604_{\pm\ 0.0022}$ | $0.0275_{\pm\ 0.0014}$ | $2.129_{\pm\ 0.0032}$ |
|  | 20k | $8.20_{\pm\ 0.04}$ | $0.09_{\pm\ 0.006}$ | $0.596_{\pm\ 0.0008}$ | $\mathbf{0.0258_{\pm\ 0.0003}}$ | $2.109_{\pm\ 0.0012}$ |
|  | 100k | $8.38_{\pm\ 0.13}$ | $0.18_{\pm\ 0.014}$ | $0.591_{\pm\ 0.0012}$ | $0.0269_{\pm\ 0.0004}$ | $2.100_{\pm\ 0.0011}$ |
| **MSA (Ours)** | 5k | $8.07_{\pm\ 0.11}$ | $0.11_{\pm\ 0.006}$ | $0.597_{\pm\ 0.0011}$ | $0.0274_{\pm\ 0.0011}$ | $2.125_{\pm\ 0.0070}$ |
|  | 20k | $\mathbf{7.98_{\pm\ 0.03}}$ | $\mathbf{0.08_{\pm\ 0.003}}$ | $0.589_{\pm\ 0.0013}$ | $0.0259_{\pm\ 0.0007}$ | $2.107_{\pm\ 0.0020}$ |
|  | 100k | $8.18_{\pm\ 0.14}$ | $0.10_{\pm\ 0.009}$ | $\mathbf{0.589_{\pm\ 0.0006}}$ | $0.0264_{\pm\ 0.0004}$ | $\mathbf{2.098_{\pm\ 0.0029}}$ |

Table 3: Evaluation of the wind nowcasting task according to standard weather metrics, which are described in Appendix H of the supplementary material. The optimal value of the metric is indicated in the parenthesis. MSA is the best model overall, with the lowest absolute error, a near-zero systematical bias, and output values that have a similar dispersion to GEN. We added the results of the GKA model for comparison as it is the best-performing model in the original paper.

| Model | RMSE ($\downarrow$) | $\theta$ MAE ($\downarrow$) | $r$ MAE ($\downarrow$) | Relative BIAS$_x$ (0.0) | Relative BIAS$_y$ (0.0) | rSTD (1.0) | NSE ($\uparrow$) |
|---|---|---|---|---|---|---|---|
| **CNP** | $10.99_{\pm\ 0.75}$ | $25.55_{\pm\ 1.22}$ | $9.22_{\pm\ 0.33}$ | $0.00_{\pm\ 0.09}$ | $-1.09_{\pm\ 0.03}$ | $1.25_{\pm\ 0.07}$ | $-0.23_{\pm\ 0.01}$ |
| **GEN** | $8.97_{\pm\ 0.06}$ | $22.56_{\pm\ 0.77}$ | $6.97_{\pm\ 0.05}$ | $-0.02_{\pm\ 0.03}$ | $-0.97_{\pm\ 0.21}$ | $\mathbf{1.09_{\pm\ 0.07}}$ | $0.25_{\pm\ 0.02}$ |
| **GKA** | $8.44_{\pm\ 0.01}$ | $21.89_{\pm\ 0.02}$ | $6.65_{\pm\ 0.02}$ | $-0.02_{\pm\ 0.00}$ | $-1.78_{\pm\ 0.02}$ | $1.13_{\pm\ 0.00}$ | $0.31_{\pm\ 0.01}$ |
| **TFS (Ours)** | $7.99_{\pm\ 0.15}$ | $22.17_{\pm\ 1.20}$ | $6.48_{\pm\ 0.50}$ | $0.08_{\pm\ 0.10}$ | $-2.21_{\pm\ 2.67}$ | $1.17_{\pm\ 0.04}$ | $0.43_{\pm\ 0.08}$ |
| **MSA (Ours)** | $\mathbf{7.36_{\pm\ 0.06}}$ | $\mathbf{20.48_{\pm\ 0.48}}$ | $\mathbf{5.67_{\pm\ 0.11}}$ | $\mathbf{0.00_{\pm\ 0.02}}$ | $\mathbf{-0.04_{\pm\ 0.64}}$ | $1.09_{\pm\ 0.02}$ | $\mathbf{0.55_{\pm\ 0.05}}$ |

in this setup. We also observe that the GKA model performs well for short time horizons when most of the information in the context is still up-to-date. However, as the time horizon increases, the GKA model's lack of flexibility is more apparent, and GEN is more competitive.

For the Heat Diffusion Experiment, we utilize the data set introduced in (Alet et al., 2019), derived from a Poisson Equation solver. The data set consists of context measurements in the unit square corresponding to sink or source points, as well as points on the boundaries. The targets correspond to irregularly sampled heat measurements in the unit cube. Our approach offers significant performance improvements, reducing the root mean square error (RMSE) from 0.12 to 0.08, (MSE reduction of 0.016 to 0.007, in terms of the original metric) as measured against the ground truth [Table 2].

For the Fluid Flow Experiment, both data sets are derived from (Li et al., 2021), subsampled irregularly in space. In both cases, MSA outperforms the alternative [Table 2]. In the Darcy Flow equation, the TFS model with 20k parameters exhibits the best performance, but this task proved to be relatively easier, and we hypothesize that the MSA model could not fully exploit this specific setup. However, it is worth mentioning that the performance of the MSA model was within a standard deviation of the TFS model.

We conducted a Two-Day Weather Forecasting Experiment utilizing ERA5 data set measurements. The data set consists of irregularly sampled measurements of seven quantities, including wind speed at different altitudes, heat, and cloud cover. Our goal is to predict wind conditions at 100 meters two days ahead based

on these measurements. MSA demonstrates its effectiveness in capturing the temporal and spatial patterns of weather conditions, enabling accurate predictions [Table 2].

To summarize, our experiments encompass a range of tasks including high-altitude wind nowcasting, heat diffusion, fluid modeling, and two-day weather forecasting. Across these diverse tasks and data sets, our proposed model consistently outperforms baseline models, showcasing their efficacy in capturing complex temporal and spatial patterns.

## 5 Understanding Failure Modes

We examine the limitations of CNP and GEN latent representation for encoding a context. Specifically, we focus on the bottleneck effect that arises in CNP from using a single vector to encode the entire context, resulting in an underfitting problem (Garnelo et al., 2018; Kim et al., 2019), and that applies similarly to GEN. To highlight this issue, we propose three simple experiments. (1) We show in which case baselines are not able to pass information in the context that they use for conditioning, and why MSA and TFS are not suffering from this problem. (2) We show that maintaining independent latent representation helped to the correct attribution of perturbations. (3) We show that this improved latent representation leads to better error correction.

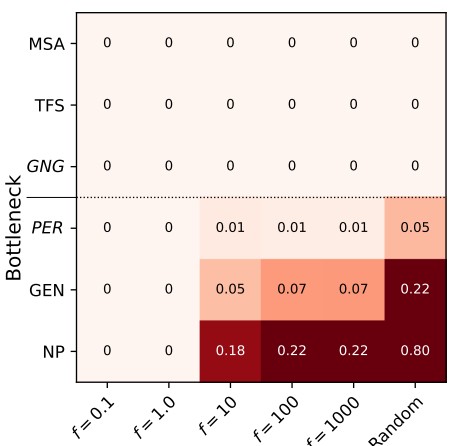

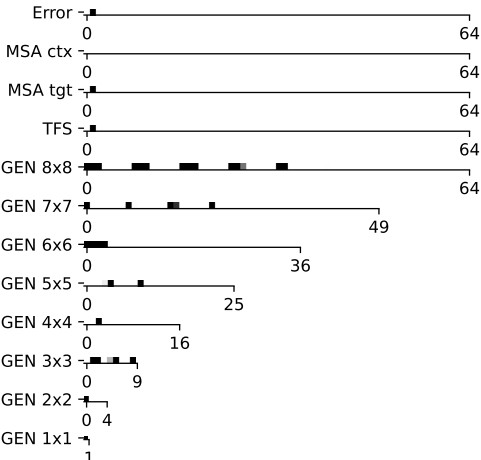

Figure 4: The results of the information retrieval experiment, evaluated using MSE, are considered satisfactory if the MSE is below 0.01. The first three rows depict models without bottlenecks. The x-axis represents datasets organized by increasing frequency, with 'Random' as the extreme case where the context value is independent of its position. Models with bottlenecks are sufficient when the learned function varies minimally in space. Models in italics denote hybrid architectures: *GNG* represents 'GEN No Graph', maintaining a latent measure per context, and *PER* indicates a transformer with a perceiver layer (Jaegle et al., 2021), introducing a bottleneck.

Figure 5: Gradients on the last layer of the encoder corresponding to an artificial error of $\epsilon = 10.0$ added to the second output. MSA maintains independent latent representation and gradients are exclusively non-zero for the latent associated with the error. We compare it to different GEN models each initialized with a graph corresponding to a regular grid of size $i \times i$ with $i \in \{1, \ldots, 8\}$. Due to the bottleneck effect, the gradients corresponding to one error are propagated across different latent vectors for GEN. Even when there are enough latents (GEN $8 \times 8$), GEN still disperse attribution because their distance-based conditioning that does not allow for a one-to-one mapping between targets and latents.

### 5.1 Capacity of Passing Information from Context to Targets

Every model considered in this work encodes context information differently. Ideally, each should be capable of using every measure in their context efficiently. We will see that excessive bottlenecking in the latent space can make this difficult or impossible.

To demonstrate this result, we design a simple experiment in which each model encodes a set of 64 measures $(\mathbf{x}_c, \mathbf{y}_c)$, and is then tasked with duplicating the corresponding $\mathbf{y}_t = \mathbf{y}_c$ given the $\mathbf{x}_t = \mathbf{x}_c$. The training and validation set have respectively 10 000 and 1 000 pairs of sets of 64 examples. It is worth noting that the models have access to all the information they need to solve the task with near-zero MSE. We conducted several experiments, starting by randomly sampling 2D context positions $\mathbf{x}_c = (x, y)$ from a Gaussian distribution and computing the associated deterministic smooth function:

$$\mathbf{y}_c = \sin(\pi f x) \cos(\pi f y) \in \mathbb{R} \tag{17}$$

where $f$ is a frequency parameter that governs problem difficulty. The higher $f$ is, the more difficult the function becomes, as local information becomes less informative about the output. We also consider as a harder problem to sample $\mathbf{y}_c$ randomly and independently from the position.

The results of this experiment, as shown in Figure 4, indicate that the CNP and GEN models are less effective in learning this task at higher frequencies. This inefficiency is primarily due to a phenomenon we define as a 'bottleneck': a situation where a single latent variable is responsible for representing two distinct context measurements. This bottleneck impedes the models' ability to distinguish and retrieve the correct target value. In contrast, models with independent latent representations, like MSA, are not subject to this limitation and thus demonstrate superior performance in learning the task. To ensure that the effect arises from a bottleneck in the architectures, we created two hybrid models, trying to stay as close as possible to the original models but removing the bottleneck from GEN and adding one for transformer-based models. We call the first one GNG (for *GEN No Graph*), which adapts a GEN and instead of relying on a common graph, creates one based on the measure position with one node per measure. Edges are artificially added between neighboring measures which serve as the base structure for $L$ steps of message-passing. This latent representation is computationally expensive as it requires the creation of a graph per set of measurements, but it does not create a bottleneck in the latent representation. We found that GNG is indeed able to learn the task at hand. We then followed the reverse approach and artificially added a bottleneck in the latent representation of attention-based models by using Perceiver Layer (Jaegle et al., 2021) with $P$ learned latent vectors instead of the standard self-attention in the transformer encoder (and call the resultant model *PER*). When $P$ is smaller than the number of context measurements, it creates a bottleneck and *PER* does not succeed in learning the task. If the underlying space is smooth enough, GEN, CNP, and *PER* are capable of reaching perfect accuracy on this task as they can rely on neighboring values to retrieve the correct information. This experiment demonstrates that MSA and TFS can use their independent latent representations to efficiently retrieve context information regardless of the level of discontinuity in the underlying space, while models with bottlenecks, such as CNP and GEN, are limited in this regard and perform better when the underlying space is smooth.

### 5.2 Improving Error Correction

In the following analysis, we explore how independent latent representations can enhance error correction during training.

To do so, we conduct an ablation study focusing on the hypothesis that maintaining independent representations is beneficial during training. We test this hypothesis by restricting the ablation to a straightforward scenario, where the models have perfect accuracy on all their outputs but one. To do so we start by pretraining the models to zero error on a validation set in a smooth case ($f = 1.0$). We then pass a fixed random validation batch to the different models and consider this output as the target for the experiment. We then deliberately add a perturbation $\epsilon = 10.0$ to one of the target values and backpropagate the error through the model, observing how the models adapt to this perturbation and the consequent impact on their latent variables.

In Figure 5, we present the norm of gradients at the last encoder layer of the latent variables for different models, specifically MSA, TFS, and GEN, when correcting an error during training. It shows how MSA and TFS models experience a non-zero gradient only in a single latent variable, whereas the GEN model exhibits changes across multiple latent variables. This disparity suggests that models like MSA and TFS, which maintain one latent variable per output, are less constrained during training. They need only to adjust the specific latent corresponding to the error without affecting others. Conversely, models with entangled representations, as seen in GEN, must manage the dual task of correcting the error while ensuring other dependent outputs remain unchanged.

We think that the additional constraints might lead to slower training and to test that hypothesis, we measure the number of backpropagation passes required to return to zero error on the validation set [Figure 6]. We see that as the number of latent variables reduces, it leads to an increase in the number of steps required to correct the error. This result

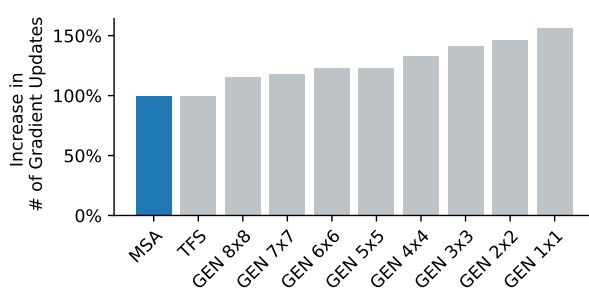

Figure 6: Comparison of the number of gradient updates required to correct an artificial error, with respect to MSA (lower is better). The y-axis represents the increase in percentage in the number of steps required to reach a perfect accuracy with respect to MSA. We compared MSA to different GEN each initialized with a graph corresponding to a regular grid of size $i \times i$ with $i \in \{1, \ldots, 8\}$. It can be observed that all GEN take more steps to correct the same mistake, and the more entangled the latent representation is, the more time it requires to correct the problem.

suggests that the entanglement of latent variables in GEN can lead to slower training and that maintaining independent latent representations can improve error correction. We make the hypothesis that this scenario extends to general training dynamics, proposing that models with independent latent representations can correct errors more efficiently due to their reduced constraints.

### 5.3 Encoding scheme

In this section, we evaluate the novel encoding scheme presented in Section 3.2, and we present the results in Table 4. We found that it reduces the RMSE from 8.47 to 7.98 in the wind nowcasting task and enables the MSA model to achieve the best performance with an RMSE of 0.08 for the Poisson Equation. Sharing the same mapping for positions is the appropriate inductive bias for encoding positions, as it eliminates the need to learn the same transformation twice. Since our data is irregularly sampled in space, the positioning of measurements and target positions significantly influence the prediction, as demonstrated in additional experiments, Appendices I and J in the supplementary material. We think that sharing the position mapping can link information from the context and target positions, which helps the model to understand better how the space is shaped.

## 6 Conclusion

In this work, we introduced an attention-based model to handle the challenges of wind nowcasting data. We demonstrated that the proposed attention-based model was able to reach the best performance for high-altitude wind prediction and other dynamical systems, such as weather forecasting, heat diffusion, and fluid dynamics when working with data irregularly sampled in space. We then explained why attention-based models were capable of outperforming other models on that task and provided an in-depth examination of the differences between models, providing explanations for the impact of design choices such as latent representation bottlenecks on the final performance of the trained models.

Our work builds upon well-established attention models, which have demonstrated their versatility and efficacy in various domains. Although the core model is essentially a vanilla transformer, our architecture required careful adaptation to suit our specific requirements. We designed our model to be set-to-set rather than sequence-to-sequence, handling data in a non-causal and non-autoregressive manner, and generating continuous

Table 4: Relative performance improvement with shared encoding for position across various models. This improvement is calculated as $\frac{\text{Error}_{\text{shared}}}{\text{Error}_{\text{no shared}}}$, where $\text{Error}_{\text{no shared}}$ represents the mean error across sizes for models without shared encoding and $\text{Error}_{\text{shared}}$ is the mean error across sizes for models utilizing shared encoding. Detailed values are available in Table 5. Scores exceeding 100% indicate superior performance without shared encoding. Generally, shared encoding enhances model efficiency, typically by a small margin, but significantly in some cases like the Poisson Equation. The shared encoding approach is likely advantageous as an inductive bias for position encoding. By naturally treating the positions of context and targets similarly, it can effectively use this information to develop more robust representations.

| Architecture | Wind Nowcasting | Poisson Equation | Navier-Stokes Equation | Darcy Flow Equation | ERA5 |
|---|---|---|---|---|---|
| CNP | 100.7% | 142.1% | 154.1% | 193.0% | 100.6% |
| GEN | 95.2% | 63.5% | 101.9% | 77.9% | 101.3% |
| TFS (Ours, baseline) | 95.3% | 79.5% | 99.5% | 99.3% | 101.0% |
| MSA (Ours) | 93.7% | 24.8% | 99.4% | 97.2% | 100.9% |

values for regression. The success of influential models like BERT (Devlin et al., 2019), GPT (Radford et al., 2018), ViT (Dosovitskiy et al., 2020), and Whisper (Radford et al., 2022), also closely resembles the original implementation by Vaswani et al. (2017), which further supports the effectiveness of the transformer framework across different tasks and domains.

Finally, our model's scalability is currently limited by its quadratic complexity in the context size. Although this limitation does not pose a problem in our particular use cases, it can impede the scaling of applications. This is a significant challenge that affects all transformer-based models and has garnered considerable attention. Recent developments to tackle this challenge include flash-attention (Dao et al., 2022), efficient transformers (Katharopoulos et al., 2020), and quantization techniques (Dettmers et al., 2022), which can address this problem, enhancing the feasibility of our approach for large-scale applications.

**Acknowledgement** Arnaud Pannatier was supported by SkySoft ATM for the project "MALAT: Machine Learning for Air Traffic" and the Swiss Innovation Agency Innosuisse under grant number 32432.1 IP-ICT. – Kyle Matoba was supported by the Swiss National Science Foundation under grant number FNS-188758 "CORTI".

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

## A   Detailed results

Here, we present a breakdown of the detailed results for the different models with and without shared positional encoding, and presented in terms of the specific metrics reported in their original study. In Table 2 of the main paper, we provide the results for GEN and CNP based on their original implementations, which do not involve sharing the position encoding and we translated all metrics to RMSE. For TFS and MSA, we include the results obtained with shared position encoding. To ensure comprehensiveness and to differentiate the enhanced performance attributed to the latent representation from that resulting from the improved data encoding approach, we also apply the shared encoding technique to GEN and CNP.

In most cases, we observe that employing a shared encoding for position tends to improve the performance of all models, except for CNP where it seems to decrease the overall performance in some cases. Additionally, it seems that MSA surpasses its competitors in the majority of instances, consistently exhibiting superior performance across various data sizes. Moreover, when the shared encoding for positions fails to achieve the optimal performance, the model's performance is generally comparable to that without shared encoding, often falling within a standard deviation range. We also give in Table 6 a detailed breakdown of the results for the training set.

## B   Dataset descriptions

### B.1   Wind nowcasting

The wind nowcasting experiment uses the same dataset as (Pannatier et al., 2021). It is available at: `https://zenodo.org/record/5074237`. It is an important task for ATC as it involves the crucial prediction of high-altitude winds using real-time data transmitted by airplanes. This prediction is essential for ensuring efficient airspace management the airspace. It is important to note that in the vertical direction ($z$), the majority of wind measurements are typically taken at elevated altitudes, specifically between 4,000 and 12,000 meters. Regarding the horizontal dimensions, the airspace extends over 600 km from north to south and approximately 500 km from east to west. As ground-based measurements are not accessible at these heights, our dependence lies on the measurements gathered directly from airplanes. As planes do not record the wind in the $z$ direction, the input space corresponds to wind speed in the $x, y$ plane $\mathbb{I} \subseteq \mathbb{R}^2$, and the output space is the wind speed measured later, $\mathbb{O} \subseteq \mathbb{R}^2$. Here the underlying space is European airspace represented as $\mathbb{X} \subseteq \mathbb{R}^3$ In this particular setup, the models need to extrapolate in time, based on a set of the last measurements.

Airplanes measure wind speed with a sampling frequency of four seconds. We split the dataset into time slices, where each slice contains one minute of data, such that each time slice contains between 50 and 1500 data points. We tried different time intervals but noticed that having a longer time slice did not improve the quality of the forecasts. The objective for the different models is to output a prediction of the wind at different query points 30 minutes later. We evaluate performance using the Root Mean Square Error (RMSE) metric, as in previous work.

### B.2   Poisson Equation

This experiment uses the same dataset as (Alet et al., 2019). It is available at `https://github.com/FerranAlet/graph_element_networks/tree/master/data`. The Poisson equation models the heat diffusions over the unit square with sources represented by $\phi(x, y) \in \mathbb{R}$ and fixed boundary conditions $\omega \in \mathbb{R}$. The equation is given by:

Table 5: Results of the High-Altitude Wind Nowcasting, Poisson, Navier-Stokes and Darcy Flow equation and the weather forecasting task. Each model ran for respectively 10, 2000, 1000, 100, and 100 epochs on an NVIDIA GeForce GTX 1080 Ti. The low number of epochs for wind nowcasting is due to the amount of data which is considerably larger than in the other experiments. The standard deviation is computed over 3 runs. We choose the configuration of the models so that every model has a comparable number of parameters. We underlying the best models for each size and we put in bold the best model overall. In this table, we kept the results in the same metric as originally reported. The first half of the table corresponds to the case without the novel encoding scheme, as opposed to the second one. The last part of the table reports the results of GeoFNO another interesting baseline, which we discussed in detail in Appendix D. In Table 2 of the main paper, we recomputed all metrics in terms of Root Mean Square Error (RMSE).

| Validation Model | Size | Wind Nowcasting RMSE ($\downarrow$) | Poisson Equation MSE ($\downarrow$) | Navier-Stokes Equation MSE ($\downarrow$) | Darcy Flow Equation RMSE ($\downarrow$) E-4 | ERA 5 MSE ($\downarrow$) |
|---|---|---|---|---|---|---|
| | | | *Default encoding for positions* | | | |
| CNP | 5k | $10.19 \pm 0.21$ | $.130 \pm .0134$ | $.492 \pm .0033$ | $9.65 \pm 0.47$ | $4.48 \pm 0.01$ |
| | 20k | $10.11 \pm 0.20$ | $.105 \pm .0012$ | $.452 \pm .0015$ | $8.71 \pm 0.11$ | $4.44 \pm 0.00$ |
| | 100k | $10.20 \pm 0.26$ | $.105 \pm .0024$ | $.430 \pm .0010$ | $8.17 \pm 0.06$ | $4.43 \pm 0.01$ |
| GEN | 5k | $9.84 \pm 2.92$ | $.017 \pm .0011$ | $.365 \pm .0012$ | $9.25 \pm 0.18$ | $4.47 \pm 0.02$ |
| | 20k | $9.24 \pm 0.35$ | $.020 \pm .0054$ | $.359 \pm .0007$ | $8.74 \pm 0.09$ | $4.44 \pm 0.00$ |
| | 100k | $9.23 \pm 0.44$ | $.048 \pm .0389$ | $.355 \pm .0006$ | $8.66 \pm 0.09$ | $4.46 \pm 0.00$ |
| TFS (Ours) | 5k | $8.75 \pm 0.14$ | $.055 \pm .0248$ | $.367 \pm .0033$ | $7.46 \pm 0.55$ | $4.46 \pm 0.00$ |
| | 20k | $8.70 \pm 0.06$ | $.017 \pm .0014$ | $.357 \pm .0019$ | $6.69 \pm 0.14$ | $4.38 \pm 0.01$ |
| | 100k | $8.67 \pm 0.07$ | $.016 \pm .0045$ | $.350 \pm .0011$ | $7.47 \pm 0.24$ | $4.42 \pm 0.01$ |
| MSA (Ours) | 5k | $8.40 \pm 0.10$ | $.048 \pm .0186$ | $.361 \pm .0054$ | $7.74 \pm 0.48$ | $4.44 \pm 0.02$ |
| | 20k | $8.47 \pm 0.12$ | $.047 \pm .0107$ | $\mathbf{.346 \pm .0042}$ | $6.76 \pm 0.10$ | $4.39 \pm 0.01$ |
| | 100k | $8.98 \pm 0.22$ | $.030 \pm .0081$ | $.350 \pm .0015$ | $7.33 \pm 0.22$ | $4.41 \pm 0.01$ |
| | | | *Sharing encoding for positions* | | | |
| CNP | 5k | $10.33 \pm 0.19$ | $.165 \pm .0023$ | $.706 \pm 0.0003$ | $17.26 \pm 0.04$ | $4.53 \pm 0.01$ |
| | 20k | $10.12 \pm 0.21$ | $.160 \pm .0011$ | $.706 \pm 0.0001$ | $17.06 \pm 0.06$ | $4.47 \pm 0.00$ |
| | 100k | $10.26 \pm 0.24$ | $.158 \pm .0006$ | $.705 \pm 0.0001$ | $16.87 \pm 0.02$ | $4.43 \pm 0.01$ |
| GEN | 5k | $8.79 \pm 0.23$ | $.017 \pm .0023$ | $.372 \pm 0.0008$ | $6.83 \pm 0.17$ | $4.54 \pm 0.01$ |
| | 20k | $9.08 \pm 0.47$ | $.018 \pm .0036$ | $.366 \pm 0.0000$ | $6.75 \pm 0.05$ | $4.50 \pm 0.01$ |
| | 100k | $9.07 \pm 0.00$ | $.019 \pm .0008$ | $.361 \pm 0.0000$ | $7.19 \pm 0.12$ | $4.50 \pm 0.01$ |
| TFS (Ours) | 5k | $8.30 \pm 0.03$ | $.025 \pm .0121$ | $.365 \pm 0.0026$ | $7.58 \pm 0.84$ | $4.53 \pm 0.01$ |
| | 20k | $8.20 \pm 0.04$ | $.010 \pm .0013$ | $.355 \pm 0.0009$ | $\mathbf{6.64 \pm 0.14}$ | $4.45 \pm 0.01$ |
| | 100k | $8.38 \pm 0.13$ | $.035 \pm .0054$ | $.349 \pm 0.0015$ | $7.25 \pm 0.19$ | $4.41 \pm 0.00$ |
| MSA (Ours) | 5k | $8.07 \pm 0.11$ | $.014 \pm .0014$ | $.357 \pm 0.0013$ | $7.51 \pm 0.61$ | $4.52 \pm 0.03$ |
| | 20k | $\mathbf{7.98 \pm 0.03}$ | $\mathbf{.007 \pm .0004}$ | $.347 \pm 0.0016$ | $6.74 \pm 0.38$ | $4.44 \pm 0.01$ |
| | 100k | $8.18 \pm 0.14$ | $.010 \pm .0017$ | $.347 \pm 0.0008$ | $6.97 \pm 0.24$ | $4.40 \pm 0.01$ |
| | | | *Different encoding scheme* | | | |
| GeoFNO | 5k | $11.38 \pm 0.34$ | $.030 \pm .0086$ | $.532 \pm .0064$ | $8.46 \pm 0.11$ | $4.43 \pm 0.02$ |
| | 20k | $11.31 \pm 0.40$ | $.035 \pm .0073$ | $.473 \pm .0103$ | $8.16 \pm 0.21$ | $4.41 \pm 0.01$ |
| | 100k | $11.11 \pm 0.47$ | $.035 \pm .0226$ | $.429 \pm .0060$ | $7.92 \pm 0.10$ | $4.41 \pm 0.01$ |
| | 1.5M | $11.28 \pm 0.40$ | $.014 \pm .0035$ | $.402 \pm .0029$ | $8.22 \pm 0.02$ | $4.42 \pm 0.01$ |

$$\begin{cases} \Delta\phi(x,y) = \psi(x,y) & \text{if } (x,y) \in (0,1)^2 \\ \phi(x,y) = \omega & \text{if } (x,y) \in \partial[0,1]^2 \end{cases} \tag{18}$$

It should be noted that the boundary constant and sources function $\phi(x,y)$ conditions can change for each sample.

The dataset used in (Alet et al., 2019) uses three dimensions for the context values $\mathbf{y}_c \in \mathbb{I} \subseteq \mathbb{R}^3$: either sources values $\phi(x,y) = \mu$ inside the domain encoded as $\mathbf{x}_c, \mathbf{y}_c = (x,y), (\mu, 0, 0)$ or boundary conditions $\phi(x,y) = \omega$ on the boundaries, encoded as $\mathbf{x}_c, \mathbf{y}_c = (x,y), (0, \omega, 1)$. The target space is one-dimensional $\mathbf{y}_t \in \mathbb{O} \subseteq \mathbb{R}$ and corresponds to the solution of the Poisson equation at that point. A sample is depicted in Figure 7.

We evaluate all models for three different sizes on this particular setup and find that MSA with the novel encoding scheme outperforms other models by a significant margin as can be seen in Table 2. We initialized

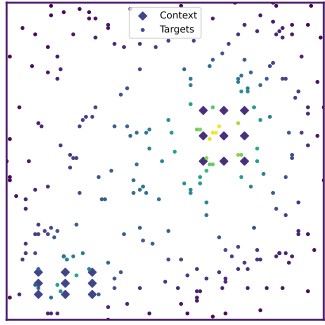

Figure 7: A sample of the Poisson equation dataset (Alet et al., 2019). ● represents the targets, The context values are comprised of points on the boundaries, and points on the sources and sink are represented by ◆ which corresponds to their thermal coefficient.

Table 6: Corresponding training metrics

| Training Model | Size | Wind Nowcasting RMSE (↓) | Poisson Equation MSE (↓) | Navier-Stokes Equation MSE (↓) | Darcy Flow Equation RMSE (↓) E-4 | ERA 5 MSE (↓) |
|---|---|---|---|---|---|---|
| | | | *Default encoding for positions* | | | |
| CNP | 5k | 11.94 ± 0.78 | .127 ± .0179 | .485 ± .0033 | 9.02 ± 0.54 | 4.33 ± 0.01 |
| | 20k | 10.19 ± 1.83 | .084 ± .0057 | .438 ± .0015 | 7.88 ± 0.16 | 4.25 ± 0.01 |
| | 100k | 10.17 ± 1.24 | .021 ± .0040 | .398 ± .0010 | 6.93 ± 0.08 | 4.17 ± 0.01 |
| GEN | 5k | 11.02 ± 3.19 | .011 ± .0006 | .362 ± .0012 | 8.49 ± 0.17 | 4.32 ± 0.02 |
| | 20k | 9.98 ± 0.76 | .006 ± .0006 | .354 ± .0007 | 7.76 ± 0.11 | 4.24 ± 0.01 |
| | 100k | 9.56 ± 0.21 | .011 ± .0122 | .339 ± .0006 | 7.01 ± 0.26 | 4.10 ± 0.01 |
| TFS (Ours) | 5k | 9.86 ± 0.21 | .022 ± .0021 | .365 ± .0033 | 6.50 ± 0.60 | 4.29 ± 0.02 |
| | 20k | 9.69 ± 0.38 | .005 ± .0008 | .353 ± .0019 | 5.48 ± 0.10 | 4.08 ± 0.03 |
| | 100k | 9.55 ± 0.19 | .001 ± .0001 | .314 ± .0011 | 4.45 ± 0.17 | 3.78 ± 0.01 |
| MSA (Ours) | 5k | 8.86 ± 0.01 | .022 ± .0062 | .359 ± .0057 | 6.91 ± 0.50 | 4.28 ± 0.02 |
| | 20k | 7.94 ± 0.03 | .005 ± .0012 | .341 ± .0042 | 5.72 ± 0.11 | 4.13 ± 0.01 |
| | 100k | 6.67 ± 0.02 | .000 ± .0001 | .310 ± .0015 | 4.91 ± 0.17 | 4.19 ± 0.01 |
| | | | *Sharing encoding for positions* | | | |
| CNP | 5k | 10.41 ± 0.03 | .154 ± .0010 | .703 ± .0001 | 16.94 ± 0.06 | 4.38 ± 0.00 |
| | 20k | 9.48 ± 0.05 | .133 ± .0004 | .700 ± .0001 | 16.68 ± 0.10 | 4.30 ± 0.01 |
| | 100k | 8.60 ± 0.05 | .107 ± .0008 | .696 ± .0001 | 16.36 ± 0.02 | 4.23 ± 0.01 |
| GEN | 5k | 10.04 ± 0.13 | .005 ± .0004 | .369 ± .0007 | 5.93 ± 0.15 | 4.42 ± 0.01 |
| | 20k | 9.75 ± 0.09 | .003 ± .0007 | .362 ± .0002 | 5.65 ± 0.06 | 4.36 ± 0.01 |
| | 100k | 9.84 ± 0.02 | .001 ± .0000 | .345 ± .0005 | 4.97 ± 0.22 | 4.34 ± 0.02 |
| TFS (Ours) | 5k | 8.78 ± 0.02 | .009 ± .0015 | .364 ± .0024 | 6.74 ± 0.93 | 4.41 ± 0.01 |
| | 20k | 7.94 ± 0.03 | .002 ± .0002 | .352 ± .0012 | 5.56 ± 0.10 | 4.26 ± 0.01 |
| | 100k | 6.57 ± 0.02 | .000 ± .0000 | .312 ± .0006 | 4.51 ± 0.24 | 4.09 ± 0.01 |
| MSA (Ours) | 5k | 8.54 ± 0.03 | .008 ± .0024 | .355 ± .0013 | 6.74 ± 0.61 | 4.38 ± 0.03 |
| | 20k | 7.73 ± 0.02 | .002 ± .0003 | .342 ± .0014 | 5.79 ± 0.34 | 4.25 ± 0.01 |
| | 100k | 6.58 ± 0.05 | .000 ± .0000 | .310 ± .0008 | 4.76 ± 0.12 | 4.14 ± 0.02 |
| | | | *Different encoding scheme* | | | |
| GeoFNO | 5k | 11.68 ± 0.30 | .008 ± .0010 | .526 ± .0070 | 7.56 ± 0.13 | 4.18 ± 0.02 |
| | 20k | 11.56 ± 0.20 | .003 ± .0005 | .467 ± .0100 | 7.02 ± 0.32 | 4.07 ± 0.03 |
| | 100k | 10.72 ± 0.27 | .001 ± .0001 | .421 ± .0068 | 5.87 ± 0.45 | 3.75 ± 0.07 |
| | 1.5M | 10.30 ± 0.93 | .000 ± .0002 | .363 ± .0192 | 2.56 ± 0.51 | 3.00 ± 0.26 |

GEN with a $7 \times 7$ regularly initialized grid on the $[0,1]^2$ as in the original work (Alet et al., 2019).

## B.3 Navier-Stokes Equation

Similarly to the darcy flow task, we adapted the Navier-Stokes equation as described in (Li et al., 2021) to an irregular setup.

The Navier-Stokes equation describes a real fluid and is described with the following PDE :

$$\begin{cases} \partial_t w(x,t) + u(x,t) \cdot \nabla w(x,t) = \nu \Delta w(x,t) + f(x), x \in (0,1)^2 & \text{if } t \in (0,T] \\ \nabla \cdot u(x,t) = 0, x \in (0,1)^2 & \text{if } t \in [0,T] \\ w(x,0) = w_0(x) & \text{if } x \in (0,1)^2 \end{cases} \tag{19}$$

For more detailed information regarding the notation, please refer to the original work by Li et al. (Li et al., 2021), specifically section 5.3.

In this study, our objective is to forecast the future state of the vorticity quantity, specifically 50 timesteps ahead, based on measurements of vorticity at different spatial locations. This approach differs from the original work, where the model was provided with ten initial vorticity grids and required to predict the complete system evolution.

To create our dataset, we subsampled the complete dataset, which consisted of the evolution of the Navier-Stokes equation solved by a numerical solver. The full dataset had dimensions of $1000 \times 1024 \times 1024$.

For our purposes, we selected pairs of slices that were separated by 50 timesteps and performed spatial subsampling. We took 64 context measurements and 256 targets as our subsampled data points. This experiment adapts the dataset available in (Li et al., 2021). It is available at `https://github.com/neural-operator/fourier_neural_operator` and can be processed with the code given to rearrange it in the irregular setup. We present a sample of the dataset in Figure 9.

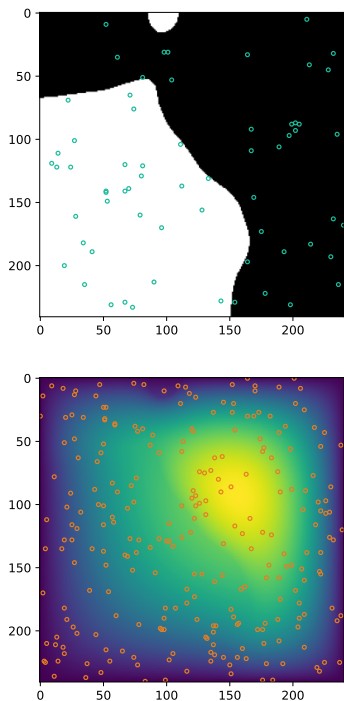

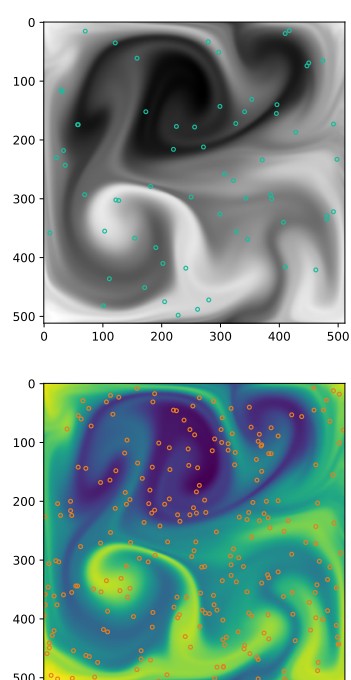

Figure 8: A sample of the Darcy Flow dataset (Li et al., 2021). Blue dots correspond to the context set and orange dots to the targets.

Figure 9: A sample of the Navier-Stokes dataset (Li et al., 2021). Blue dots correspond to the context set and orange dots to the targets.

### B.4 Darcy Flow

We evaluated the performance of various models in solving the Darcy Flow equation on the unit grid with null boundary conditions, as described in (Li et al., 2021). Specifically, we aimed to predict the value of the function $u$, given the diffusion function $a$, with the two related implicitly through the PDE given by:

$$\begin{cases} -\nabla \cdot (a(x)\nabla u(x)) = 1 & \text{if } x \in (0,1)^2 \\ u(x) = 0 & \text{if } x \in \partial[0,1]^2. \end{cases} \tag{20}$$

The dataset used in our study is adapted from that used in (Li et al., 2021), and originally generated by a traditional high-resolution PDE solver. The dataset consists of a $1024 \times 1024$ grid, which was subsampled uniformly at random and arranged into context-target pairs. The context is comprised of the evaluations of the diffusion coefficient: $(\mathbf{x}_c, \mathbf{y}_c) = ((x,y), a(x,y))$, and the target is the solution of the Darcy Flow equation at a position, $(\mathbf{x}_t, \mathbf{y}_t) = ((x,y), u(x,y))$.

The results are presented in Table 2, they are coherent with the rest of the experiment and show that the MSA and TFS models outperform all competing models. We used the same initialization for GEN as for the Poisson Equation.

This experiment adapts the dataset available in (Li et al., 2021). It is available at `https://github.com/neural-operator/fourier_neural_operator` and can be processed with the code provided to rearrange it in the irregular setup.

### B.5 Two-Days Weather Forecasting

In this task, we want to evaluate our models on the task of two-day weather forecasting based on irregularly sampled data in space. The requested data collection description focuses on climate reanalysis from the

| Dataset | dim($\mathbb{X}$) | dim($\mathbb{I}$) | dim($\mathbb{O}$) | # Train. points | # Val. points | URL |
|---|---|---|---|---|---|---|
| **Wind Nowcasting** | 3 | 2 | 2 | 26.956.857 | 927.906 | (Pannatier et al., 2021) Dataset page |
| **Poisson Equation** | 2 | 3 | 1 | 409.600 | 102.400 | (Alet et al., 2019) Github Repository |
| **Navier-Stokes Equation** | 2 | 1 | 1 | 48.883.20 | 11.673.600 | (Li et al., 2021) Github Repository |
| **Darcy Flow Equation** | 2 | 1 | 1 | 409.600 | 102.400 | (Li et al., 2021) Github Repository |
| **ERA5** | 2 | 7 | 2 | 8.648.704 | 2.107.392 | (Hersbach et al., 2023) Dataset Store |
| **Random** | 2 | 1 | 1 | 640.000 | 64.000 | Randomly generated |
| **Sine** | 2 | 1 | 1 | 640.000 | 64.000 | Randomly generated |

Table 7: Description of the different datasets used in this study.

ERA5 dataset, which is available through the Copernicus Climate Data Store (CDS). The dataset contains various parameters related to wind, surface pressure, temperature, and cloud cover.

To access the data, please visit the following URL: `https://cds.climate.copernicus.eu/cdsapp#!/dataset/reanalysis-era5-single-levels?tab=form`.

We selected the following variables:

- 10m u-component of wind: This refers to the eastward wind component at a height of 10 meters above the surface.

- 10m v-component of wind: This represents the northward wind component at a height of 10 meters above the surface.

- 100m u-component of wind: This denotes the eastward wind component at a height of 100 meters above the surface.

- 100m v-component of wind: This signifies the northward wind component at a height of 100 meters above the surface.

- Surface pressure: This represents the atmospheric pressure at the Earth's surface.

- 2m temperature: This indicates the air temperature at a height of 2 meters above the surface.

- Total cloud cover: This refers to the fraction of the sky covered by clouds. The data collection includes measurements for all times of the day and all days available in the dataset. However, for training purposes, we selected the data from the year 2000. For validation, we use the data from the year 2010, specifically the months of January and September.

Next, we proceeded to extract the corresponding grib files. To create our dataset, we paired time slices that had a time difference of two days, equivalent to 48 timesteps since ERA5 data has a one-hour resolution over the whole world.

For the context slice, we performed subsampling at 1024 different positions and retained all seven variables mentioned earlier. As for the target slice, we subsampled it at 1024 different positions and kept only the two components of the wind at 100m.

## C   Specific Aspects of Our Approach Compared to Existing Works

In our research, we address a distinct challenge that sets our work apart from conventional mesh-based simulators (Pfaff et al., 2020; Janny et al., 2023; Li et al., 2022a). Our focus is on extrapolating sets of sparse and variably-positioned measurements, a context markedly different from the consistent mesh structures employed in traditional approaches. Unlike mesh-based models where values are accessible at fixed points

on an irregular mesh, our method tackles scenarios where the quantity and locations of measurements can change dynamically from one data set to another. This unique aspect of our work necessitates a departure from the conventional mesh framework. Mesh-based simulators, while effective in their domains, are not equipped to handle the sporadic and non-uniform nature of our data. For instance, in applications such as airspace monitoring, data points represent measurements taken only where planes are present, leaving vast areas without data. Our methodology, therefore, diverges fundamentally from mesh-based techniques, as it requires innovative handling of sparse and irregular data inputs, rather than relying on a fixed, uniform mesh structure.

Furthermore, our approach markedly differs from trajectory prediction models (Yuan et al., 2021; Nayakanti et al., 2023; Girgis et al., 2022). While these models excel in forecasting future points along established trajectories, our work diverges in both intent and application. Our primary concern is not the sequential prediction of trajectories but rather the interpretation and forecasting of phenomena in a spatially and temporally sparse environment. In the context of wind nowcasting, for example, our model excels at processing limited and scattered data points from multiple trajectories to generate a comprehensive forecast. This capability to assimilate sparse measurements from varied locations and synthesize them into a coherent prediction sets our work apart from traditional trajectory-focused models. These models, such as Wayformer (Nayakanti et al., 2023) and Agentformer (Yuan et al., 2021), are primarily designed for time series forecasting and trajectory completion.

## D  Comparison with GeoFNO

The Fourier Neural Operator (FNO) family (Li et al., 2021) and its variant GeoFNO (Li et al., 2022b), have emerged as a significant tool in resolving Partial Differential Equations (PDEs) across various benchmarks. Even though the GeoFNO model was specifically designed for irregular meshes, setting it apart from other models evaluated, it is recognized for its adept handling of irregular geometries and flexibility in adapting to different types of Partial Differential Equations (PDEs), making it a strong baseline for our experiments.

We evaluated GeoFNO on our specific dataset and problem. We tried both the original model configuration, consisting of 1.5 million parameters, and its scaled-down versions with 5k, 20k, and 100k parameters, as we found the 1.5 million parameters to be over-parametrized. The outcomes of these tests can be found in Table 5.

Despite GeoFNO's overall robustness, we found that it encounters challenges in this particular setup. We hypothesize that the model struggled with sparse irregular data, as it needs to learn to map a set of data to a grid before applying a regular FNO layer. Sparsity, in this context, might lead to several parts of the grid being empty thus leading to information loss or dispersion. Secondly, the experiment's unique requirement of handling variable measurement positions, which is different from the usual consistent yet irregular meshing, introduced an additional layer of complexity in learning a stable mapping. These factors potentially contributed to the performance inconsistencies observed in our experiments. For instance, in the Poisson setup, where data points were more evenly distributed, GeoFNO's performance occasionally matched that of the MSA model in some randomized trial runs, as indicated in [Figure 12e]. However, in more demanding scenarios like wind nowcasting, GeoFNO consistently underperformed in comparison to both MSA and GEN models.

## E  Attention matrix for Wind Nowcasting

For all models, we can plot the norm of the input gradients to see how changing a given value would impact the prediction [Figure 10]. We see that Transformers seem to find a trade-off between taking into account the neighboring nodes and the global context.

## F  Comparison with Graph Kernel Averaging

Smart averaging strategies such as GKA are limited by design to the range of the values in their context. However, these baselines are useful as they are not prone to overfitting and in the case of time series forecasting

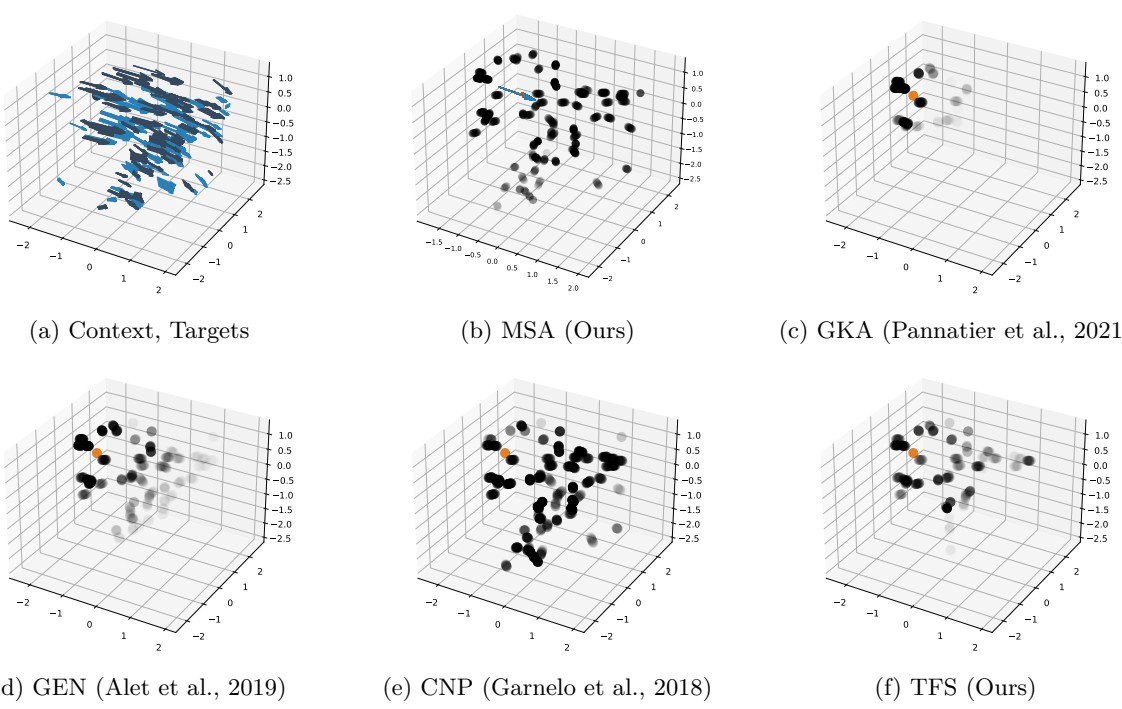

(a) Context, Targets       (b) MSA (Ours)       (c) GKA (Pannatier et al., 2021)

(d) GEN (Alet et al., 2019)       (e) CNP (Garnelo et al., 2018)       (f) TFS (Ours)

Figure 10: Displaying the importance given to points in the context to do the prediction for the different models for a given query point (orange). We use the norm of the input gradients for that purpose and highlight the context values that have the largest gradient with respect to the output. The opacity of the dots corresponds to the relative magnitude of the input gradients compared to other points in the context.

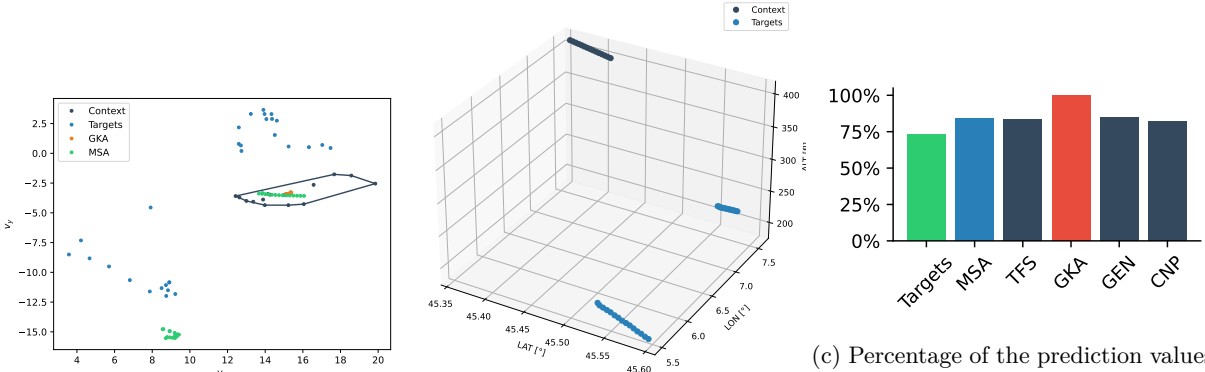

(a) Context, targets, and forecast in wind speed space.

(b) Context and targets in the 3D space.

(c) Percentage of the prediction values that are contained in the convex hull of the measurement in the context in percent.

Figure 11: Context and targets represented in wind speed and 3D space. Both the context and targets correspond to 1-minute slices of data. The targets capture wind measurements recorded 30 minutes following the context data. Additionally, the forecasts generated by the MSA and GKA models are depicted. Notably, the GKA model is confined to predicting within the convex hull of the context values, which is visually highlighted in grey. Conversely, the MSA model is unrestricted by this limitation and can make predictions beyond the convex hull of the context values.

(if the underlying variable has some persistence), the last values would in general be the best predictor of the next value. We hypothesize that over sufficiently short horizons, these methods would achieve performance similar to that from more complicated models, but when the prediction horizon rises, this design limitation would become more pronounced.

Since, for extrinsic forecasting, target points need not lie in the convex hull of context points, the greater flexibility of MSA and TFS can make correct forecasts that are impossible with GKA. Figure 11a demonstrates this fact, with a plot of the target variable, the context points, and the context convex hull in wind speed space.

But greater capacity means also more failure modes. As we saw in the metrics of Table 3 of the main paper, MSA and TFS are the only models to outperform GKA whereas other models, even if they are more flexible, fail to beat this baseline. On average 27% of the predictions were outside the convex hull of the context. In Figure 11c we analyze the percentage of measurements outside the convex hull produced by the different models. We can see that GKA is limited as 100% of the measurements lie within the convex hull whereas the other models can compensate and predict values outside it.

## G    Decoder and conditioning function

In the main paper, we presented a comparison of the key distinctions between our attention-based architectures. However, it could be that the performance differences observed could be attributed to minor variations in the model architecture design. For the sake of completeness, in this section, we provide experimental results to address this aspect by comparing the remaining differences between GEN and TFS. Specifically, we focus on analyzing the impact of the number of decoder layers and the conditioning function. It is important to note that this analysis does not apply to MSA, as it does not possess an encoder-decoder structure.

GEN and TFS differ not only in their latent representation but also on two other points: (1) the way that they access the encoded information in the decoder and (2) how much computing power is used in the decoder stack.

Table 8: Results of the ablation of the conditioning function used to combine the latents with the target position. We adapted the architecture so that they used both a distance-based conditioning function, which combines the query position with a weighted average of the nearest latents and standard cross-attention. We show in italics hybrid architectures where we had to switch the default conditioning method. For GEN and CNP replacing the conditioning method does not impact the performance, but for TFS switching to a distance-based conditioning function impacts the performance drastically.

| Model | Distance-based | Cross-attention |
|---|---|---|
| **CNP** | $.115 \pm .015$ | *$.119 \pm .014$* |
| **GEN** | $.024 \pm .004$ | *$.022 \pm .006$* |
| **TFS** | *$.042 \pm .023$* | **$.020 \pm .011$** |

Table 9: Results of the decoder-layer ablation. One difference between TFS and GEN is that by default TFS uses multiple layers in its decoder whereas GEN uses one, and delegates all processing to the encoder. We see that having multiple decoder layers helps the transformer whereas it impacts the performance of the GEN.

| | # Decoder layers | | |
|---|---|---|---|
| **Model** | **1** | **2** | **4** |
| **GEN** | $.021 \pm .002$ | $.040 \pm .012$ | $.106 \pm .004$ |
| **TFS** | $.020 \pm .011$ | $.019 \pm .005$ | **$.013 \pm .001$** |

In all three models, we tried using either a distance-based function or cross-attention to combine the target position with the encoded latents. CNP concatenates the same context vector to all target queries, which we model as equivalent to averaging the latents based on the distance in the case that there is only one latent.

Specifically, each latent is associated with a position $\mathbf{x}_l$ in the underlying space with corresponding value $\mathbf{l}_y$. Conditioning is made by averaging the latent features based on their distance with the query $\mathbf{x}_t$. Its formula is given by:

$$z = \sum_{\mathbf{l} \in \text{latents}} r(\text{dist}(\mathbf{x}_l, \mathbf{x}_t))\mathbf{l}_y \tag{21}$$

Where $r : \mathbb{R} \to [0, 1]$ is a function that maps distances to weights, and $\mathbf{x}_l$ represents the position associated with the latent nodes with value $\mathbf{l}_y$. This vector $z$ is then concatenated to the target position $\mathbf{x}_t$ and fed to the decoder which is an MLP in this case.

GEN uses this distance-based aggregation function by default whereas TFS uses cross-attention. We tried using cross-attention for CNP and GEN and using the same distance-based aggregating function for TFS using the distance between context position and target position as a reference for this scheme. For GEN and CNP using cross-attention gives approximately the same results, but using distance-based conditioning for TFS hurts the performance significantly.

We also ran ablations that created hybrid cases for both transformers and GEN and concluded that adding decoding layers helped TFS reach better performance. Adding layers to the GEN decoder drastically reduces model performance, which seems to be coherent with the fact that GEN was shown to be more prone to overfitting in the results section.

Finally, we want to highlight that MSA outperforms all the models presented in this section by a large margin Table 2.

## H   Wind nowcasting metrics

This section defines the metrics used for wind nowcasting.

**Root Mean Square Error (RMSE)** It takes the square root of the square distance between the prediction and the output. It has the advantage of having the same units as the forecasted values.

$$\text{RMSE}(\hat{t}, t) = \sqrt{\frac{\sum_k^N (\hat{t}_k - t_k)^2}{N}} \tag{22}$$

**Mean Absolute Error for angle (angle MAE)** It is interesting and sometimes more insightful to decompose the errors made by the models in their angular and norm components. This is the role of this metric and the following.

$$\text{angle MAE}(\hat{t}, t) = ||(\alpha(\hat{t}) - \alpha(t))||_{L_1} \tag{23}$$

where $\alpha \left( \vec{x} = \begin{pmatrix} x \\ y \end{pmatrix} \right) = \arctan(y, x) * \frac{360}{2\pi}$

**Mean Absolute Error for norm (norm MAE)** Following the explanation of the last paragraph:

$$\text{norm MAE}(\hat{t}, t) = \sum_k | \, || \, \hat{t}_k \, ||_2 - || \, t_k \, ||_2 \, | \tag{24}$$

**Relative Bias (rel BIAS) x,y** Additionally, we used weather metrics as in (Ghiggi et al., 2019). The relative bias measures if the considered model under or overestimates one quantity. It is defined as:

$$\text{rel BIAS}(\hat{t}, t) = \frac{\text{mean}(\hat{t}_k - t_k)}{\text{mean}(\hat{t}_k)} \tag{25}$$

**Ratio of standard deviation (rSTD)** The ratios of standard deviation indicate whether the dispersion of the output of the model matches the target distribution. It has an optimal value of 1.

$$\text{rSTD}(\hat{t}, t) = \frac{\text{std}(\hat{t})}{\text{std}(t)} \tag{26}$$

**NSE** The last domain metric used is the Nash-Sutcliffe efficiency (NSE), which compares the error of the model with the average of the target data. A score of 1 is ideal and a negative score indicates that the model was worse than the average prediction on average.

$$\text{NSE}(\hat{t}, t) = 1 - \frac{\text{MSE}(\hat{t}, t)}{\text{MSE}(t, \text{mean(t)})} \tag{27}$$

## I  Influence of the tightness of the measurements on the performances

To evaluate the influence of the tightness of the measurements on the results we designed the following experiment: We start with a pair of context $(\mathbf{x}_c, \mathbf{y}_c)$ and target points $(\mathbf{x}_t, \mathbf{y}_t)$. We pick one point from the context and restrict the context to a small number of measurements close to this point. Then, we calculate the error made by the model when it is given only this restricted context and rank them by their distance to the context's center. As the dimensions (longitude, latitude, altitude) differ, we first normalized the data along each axis. We repeat this process and average the results.

We see that the error increases with the distance to the context. Now in practice, the context is not restricted and contains points all over the space (so the mean minimum normalized distance to the target points is usually often below 1.0)

Table 10: Influence of the tightness of the measurements on the performance. We restrict the context to measurements that are close to the query point and then measure the error as a function of the distance from this query point averaged over the whole dataset. We estimate the standard deviation using three random seeds.

| Normalized Dist | MSA | TFS |
|---|---|---|
| **0.5–1.0** | $5.56 \pm 0.49$ | $7.73 \pm 0.37$ |
| **1.0–1.5** | $6.62 \pm 0.35$ | $7.35 \pm 0.53$ |
| **1.5–2.0** | $7.57 \pm 0.43$ | $7.87 \pm 0.36$ |
| **2.0–2.5** | $9.52 \pm 0.47$ | $9.48 \pm 0.66$ |
| **2.5–end** | $10.35 \pm 0.49$ | $10.53 \pm 0.61$ |

We assessed general uncertainty by training the model using various random seeds, resulting in an ensemble of models from which we can determine the standard deviation and add it to the results table of the previous experiment.

## J  Influence of the target positions

In this section, we assess the impact of the target position on the training of the MSA model.

The context data for the Poisson equation is heavily prescribed (where context points belonged to the source or sink or boundary conditions), while for the Darcy Flow equation, we constructed a similarly-shaped data set by subsampling the data set from (Alet et al., 2019) to create the context and targets. Thus, the Darcy flow test problem gives an ideal testbed for examining the influence of target positions on the training. To examine the impact of the context position, we designed an additional experiment where we sampled the context from the bottom left quadrant of the unit cube and the targets from the upper right quadrants of the unit cube. We conduct our analysis for small (5k parameters) and large (100k parameters) models. We report the test loss as a function of the percentage of the context from the upper-right quadrants (where all targets are located, the remaining from the bottom-left quadrant).

Table 11: Influence of the target position on the training. We report the test error for the MSA model at two different sizes. We devised an experiment in which we extracted the context from the bottom left quadrant of the unit cube, while the targets were sampled from the upper right quadrants of the same unit cube.

|  | 0% | 2% | 25% | 50% | 100% |
|---|---|---|---|---|---|
| **MSA 5k** | 0.14 | 0.15 | 0.14 | 0.09 | 0.03 |
| **MSA 100k** | 0.14 | 0.14 | 0.13 | 0.09 | 0.03 |

In the first column (0% of the data from the first quadrant), we see that the MSA model struggled to learn when the context and targets are disjoint. The error improves with greater overlaps between context and target, with p = 100% corresponding to the case in the paper (albeit on one-quarter of the data). Similar dynamics hold for both model sizes. We believe that this is due to the Darcy Flow simulation being highly location-dependent, as can be seen in some examples from the dataset Figure 8.

Regarding the training performance, we noticed no substantial difference in the total time to a solution, they all took approximately 1000 epochs in all cases. Larger models were always able to overfit the data.

## K  Different Message Passing Scheme

We try here different message-passing schemes even one using attention, to help us demonstrate the fact that the whole family of models that encodes the space as a single graph suffers from the same bottlenecking effect. Here are the results of the wind nowcasting task:

Table 12: Influence of the target position on the training. We report the train error for the MSA model at two different sizes. Both models were able to overfit the data.

|  | 0% | 2% | 25% | 50% | 100% |
|---|---|---|---|---|---|
| **MSA 5k** | 0.07 | 0.07 | 0.07 | 0.05 | 0.02 |
| **MSA 100k** | 0.00 | 0.00 | 0.00 | 0.00 | 0.00 |

Table 13: Performance of GEN with different message passing schemes for the wind nowcasting task. Various message-passing schemes, including those with attention, were explored using PyTorch Geometric (Fey & Lenssen, 2019). The best-performing GEN model with the default message-passing scheme is indicated with a reference line. While certain schemes showed improvements over GEN's performance, the overall performance remained relatively consistent and was still outperformed by MSA. For additional information on the different message-passing schemes and related references, please refer to the PyTorch Geometric documentation.

| Model | Score |
|---|---|
| **TransformerConv** | 9.28 |
| **GATv2Conv** | 9.34 |
| **GeneralConv** | 9.37 |
| **GATConv** | 9.77 |
| **SAGEConv** | 9.78 |
| **ARMAConv** | 9.84 |
| **TAGConv** | 9.87 |
| **SGConv** | 9.95 |
| **SuperGATConv** | 10.10 |
| **GCNConv** | 10.13 |
| **LEConv** | 10.61 |
| **GENConv** | 23.98 |

Although certain message-passing schemes enhance the performance of the GEN model, it still falls short of the performance achieved by MSA. We attribute this difference to the bottlenecking effect caused by the latent graph. For additional information on the various message-passing schemes and relevant references, please consult the PyTorch Geometric documentation available at `https://pytorch-geometric.readthedocs.io/en/latest/modules/nn.html#convolutional-layers`.

## L  Hyperparameter sensitivity analysis

In this section, we analyze the robustness of MSA to hyperparameter changes. We focus on the number of layers [Figure 12a] and the hidden dimension [Figure 12c] in the case of the Poisson equation. In the Table 5, we report three different sizes of the different models, and we tried to keep them as balanced as possible. The exact configurations of the models can be found in the config folder of the code, however, for the sake of completeness, we grid-searched over the mentioned hyperparameters while randomly sampling the learning rate.

We also did it for the width hyperparameter of GeoFNO, in the case of the Poisson equation [Figure 12e] and the wind nowcasting experiment [Figure 12g]. In the case of the Poisson experiment, the randomized trial helps in finding the best hyperparameter, leading to a model that is on par with MSA, we suspect that in this setup, the context positions are more evenly distributed in space, which might lead to a setup that does not suffer as much from the flaws listed in Appendix D, however in the case of the wind experiment, the randomized experiment did not help as much and the model is still lacking behind MSA and GEN. However, it might be possible to GeoFNO to this special case of dataset irregularly sampled in space is an interesting avenue for future work.

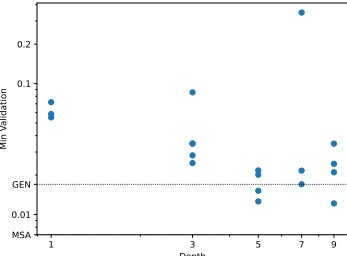

(a) Impact of the number of layers on the performance of MSA in the Poisson Experiment.

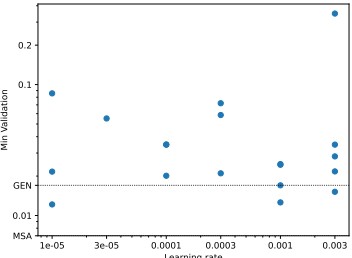

(b) Validation loss in function of the learning rate when varying the number of layers for MSA in the Poisson Experiment.

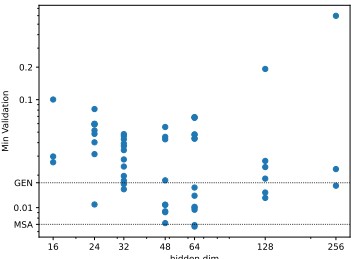

(c) Impact of the hidden dimension on the performance of MSA in the Poisson Experiment.

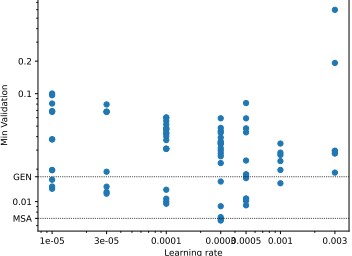

(d) Validation loss in function of the learning rate when varying the hidden dimension for MSA in the Poisson Experiment.

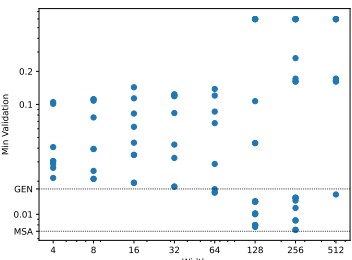

(e) Impact of the width hyperparameter on the performance of GeoFNO for the Poisson equation.

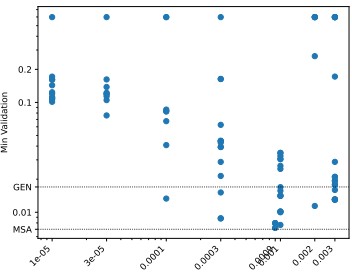

(f) Validation loss in function of the learning rate when varying the width hyperparameter for the Poisson equation.

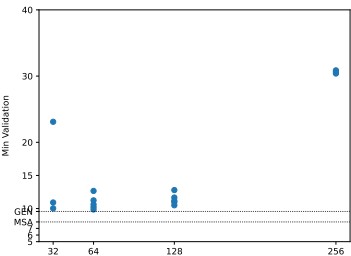

(g) Impact of the width hyperparameter on the performance of GeoFNO for the wind nowcasting task.

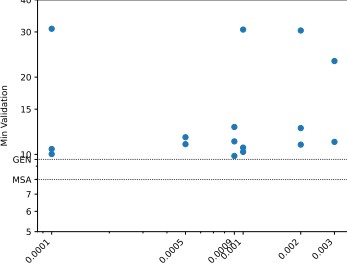

(h) Validation loss in function of the learning rate when varying the width hyperparameter for the wind nowcasting task.

Figure 12: Different hyperparameter sensitivity analysis for the Poisson equation and the wind nowcasting task.

## M   Broader Impacts

We do not anticipate any significant adverse effects on society due to this work. Generally, having an additional method for weather nowcasting may lead to an increase in computational resources required, but this is a common consequence of many deep learning systems.

We believe that enhancing the reliability of weather forecasts and dynamical systems has a more positive impact on society. By improving the accuracy and precision of these predictions, we can provide valuable information for various sectors, such as agriculture, transportation, disaster management, and overall planning and decision-making processes.

## N   Limitations

Our method, like other attention-based models, suffers from quadratic scaling. In the case of MSA, it is slightly more computationally intensive compared to TFS due to the combination of target and context in the same sequence. This results in an attention mechanism that scales with $\mathcal{O}((N_c + N_t)^2)$, which is asymptotically equivalent to the scaling of transformers, which is $\mathcal{O}(N_c^2 + N_t^2 + N_c N_t)$, but in practice introduces an additional $N_c N_t$ term. However, in all our experiments, this computational overhead did not pose a problem. We acknowledge the issue in the scaling of the model and refer to possible solutions outlined in the related work to address this challenge.

Another drawback of this study is the absence of a comparison between the models and other established models used for modeling dynamical systems on a grid, as discussed in the work by Li et al. (2021). We anticipate that our approach may not perform as effectively as competing models on grids due to two reasons. Firstly, the previously mentioned scaling issue becomes significant when dealing with larger grid sizes, such as $1024 \times 1024$ grids. Additionally, our approach lacks certain inductive biases that aid in grid-based modeling. Nonetheless, we believe that attention-based models still hold promise for grid-based systems, as demonstrated by the success of Vision Transformers (ViT) in other domains(Dosovitskiy et al., 2020). However, tokenization may be necessary for their effective implementation.

## O   Experiment and training details

In this supplementary material, we provide the code required to process the dataset and reproduce the experiments. We use Hydra as a configuration manager (Yadan, 2019) and include the precise configuration for each case. The experiments were executed multiple times on a GPU cluster, but each experiment can be conducted on a single GPU in a relatively short timeframe, ranging from a few hours to a maximum of a few days.

## P   Amount of Compute needed to replicate the experiments

Training smaller models (5k, 20k) usually takes a few hours. Training larger models (except for CNP which is considerably faster) runs in at most two GPU days. We estimate the number of GPU days to replicate all experiments for all models to be on the order of 100 GPU days.

## Q   Reproductibility

We provide the link to the dataset and the whole code base for processing it and running the experiments.

## R   Licenses

All concerned databases are openly accessible on the web and have permissive licenses, we give a link to each dataset in Table 7

