# OpenReview forum: "Inference from Real-World Sparse Measurements"
_TMLR — Accepted by TMLR_

### Review · Reviewer_hrQX · 2023-12-10

**Summary Of Contributions:**

This paper presents a simple model Multi-layer Self-Attention (MSA) to recover the complete field from sparse observations. MSA firstly employs a shared encoding layer to project the input observations and queries into deep representations and then infer the values on the query position with self-attention. Experimentally, MSA surpasses canonical Transformers, GNNs and CNPs in Wind Nowcasting and Poisson, Navier Stokes and Darcy Flow equation and the weather forecasting task.

**Audience:**

Yes

**Broader Impact Concerns:**

This paper only focuses on technical designs. Thus, there are no ethical risks and extra concerns.

**Claims And Evidence:**

Yes

**Requested Changes:**

In general, I think this paper is well-qualified. More detailed explanations and comparisons to more baselines are expected.

**Strengths And Weaknesses:**

## Strengths
-	This paper is well-written.

-	Extensive experiments and detailed analysis are provided to evaluate the model's effectiveness.

-	The experiments to test the representation disentanglement are interesting.

## Weaknesses
-	The proposed MSA model is not significantly over the canonical Transformer (TFS). Also, with Section 5.1-5.3, I can understand why Transformer-based models outperform other models. But, what about the gap between TFS and MSA? More explanations are expected. For example, why removing Decoder and shared encoder can benefit the model performance?

-	The hyperparameter Sensitivity test is missing, such as the number of layers and deep representation channels.

-	More baselines are also expected. Geo-FNO [1] can also handle the irregular mesh, which should be compared.

[1] Fourier Neural Operator with Learned Deformations for PDEs on General Geometries, arXiv 2022

---

> ### Author Response · Authors · 2024-01-12
>
> Dear Reviewer hrQX,
>
> Thank you for your review. We will start by addressing the additional baseline and then discuss the other weaknesses you mentioned.
>
> **Additional Baseline**
>
> Starting with the additional baseline, the FNO family is indeed very interesting for this problem as it is state of the art in many learning-based PDE solver benchmarks and GeoFNO seems to be applicable to the problem at hand. However, there are a few points that might hinder its performance. First there is the problem that the positions are sparse in space so learning a mapping to a grid might lead to some portion of the final grid to be empty which can create some spread out of the information. Second, as the positions of the measures change from set to set, learning a reliable mapping can be more complicated than in the case where we always have the same dense but irregular mesh.
> We adapted the original implementation to our case and tested it again at original size (1.5M parameters, which we found to be overspecified for the datasets at hand) and three smaller sizes 5k, 20k and 100k. The original implementation used a `code` variable corresponding to global parameters that we don’t have in our experiment so we removed that part.
>
> We have modified the paper to include these new results and we will upload it here once we have received the last review, following the TMRL recommendation along with the updated code and configurations files. But we list the main points here already, and we give the added results and text in the following comment.
>
> The results can be found in the main results table of the appendix, and we also added appendix D.
> Overall we found that the results were lacking behind in most of the experiments which is probably due to the reasons we listed before. We conducted more in-depth randomized trials for the poisson and the wind experiments. There we find that in some runs it was able to reach MSA’s performance in the case of the Poisson experiment, while in the case of the wind most of the runs in the randomized trial were lacking behind both MSA and GEN. We suspect that in the Poisson setup, the measurements are more evenly distributed which might explain why GEOFNO does not suffer as much as for the wind nowcasting.
>
> **MSA vs TFS**
>
> Concerning your remark on MSA vs TFS, our main hypothesis is that the lack of residual connection, caused by the cross-attention layer might explain the difference. So removing the encoder-decoder structure leads to a single path from context, targets to output which contains residual connection, allowing for a better gradient flow. Second, processing all the information in the same computational space, as opposed to processing the context and the target position independently, might help the model construct more reliable representations. We have expanded Section 3.4 to emphasize these explanations further.
>
> **Concerning Hyperparameter Sensitivity**
>
> We report three different sizes of the different models, we tried to keep them as balanced as possible. The exact configurations of the models can be found in the config folder of the code. We thought it was enough but for the sake of completeness we ran two randomised trials for the width parameter of geoFNO in the case of the wind and Poisson equation, exploring different sizes while randomising the learning rate to ensure fair comparison. We did the same for MSA for the depth, and the hidden unit size in the case of the Poisson equation and we added that part to appendix L as well.
> We'll already give the added text in the next comment.

---

> ### Author Response · Authors · 2024-01-12
>
> Dear reviewer hrQX, now that the discussion period has started, we removed the details that we previously sent here and included then in the updated version, and highlighted the changes corresponding to your remarks. In particular, we added the geoFNO baseline results in Table 4-5 a description in appendix D and the randomised trials in appendix L.

---

### Review · Reviewer_dDd7 · 2023-12-17

**Summary Of Contributions:**

This study introduces an attention-based model to address the issue of learning from irregularly sampled data. When compared with baseline models, the proposed model improves performance in a wide range of diverse tasks. Moreover, a detailed analysis is conducted to illuminate the reasons behind the superior performance of proposed models over others.

**Audience:**

Yes

**Claims And Evidence:**

No

**Requested Changes:**

- Please address the question: why does the ability of accurately storing data enhance the performance in tasks such as the wind nowcasting?
- Improve the presentation.

(for details pls refer to the weaknesses)

**Strengths And Weaknesses:**

Strengths:
- The issue of learning from irregularly sampled data is important. In cases like robotic control, the data is multi-modal, asynchronous, and irregularly sampled. It is imperative to develop an efficient approach to manage such circumstances.
- The proposed method is simple yet effective, enhancing performance across a variety of diverse tasks.
- This study earnestly addresses the challenge of comprehending the mechanisms behind their model. Through detailed experimentation, the bottleneck effect has been identified as the pivotal element impacting the model's performance.

Weaknesses:
- (Main Concern) While I appreciate the authors' endeavor to explore the failure modes, there appears to be a lack of continuity in the logical sequence: The analysis in section 5 focusing on a separate information retrieval task rather than the initial ones. Detailed experiments demonstrate the bottleneck effect and the advantages of the disentangled latent representation. The information retrieval task necessitates that the models exhibit a precise recollection of the data. Nevertheless, it is questionable that whether the initial tasks demand such a strong requirement. To be concrete, the gap in the logic is *why the ability of accurately storing data enhances the performance in tasks such as the wind nowcasting.*
  - Typically, DL models  strive to avoid memorizing all the data to prevent issues like overfitting.
  - In addition, when the model mixes two context inputs (instead of remembering them independently), the resulting fused representation could be beneficial (e.g., see works in the multimodal learning literatures)
  - Therefore, the aforementioned logical gap needs to be carefully addressed.
- Regarding the discussion in subsection 5.2:
  - the gradient-based analysis only concern the well-trained model (i.e., pre-trained to zero error), a model during training can have different behaviors. It may be too strong to use the word "independently"
  - In Fig.6 the GEN5X5 case is slightly lower than the GEN6X6 case, indicting the underlying training dynamic could be too complex to be summarized solely by the degree of representation entanglement.
- Regarding the presentation:
  - In subsection 3.1 regarding symbol notation, it would be more intuitive to adopt $x_c, y_c$ and $x_t, y_t$ and not $c_x, c_y$ or $t_x, t_y$. It is much more intuitive to use $x$ for features and $y$ for targets/labels with subscripts such as $t$ and $c$ indicating their identities.
  - From subsection 3.2 to 3.4, symbols have not been adequately defined. As an example, $\gamma(\cdot)$, which presumably denotes a FC readout module, are not explicated (Interestingly, I also note in section 5.2, $\gamma$ is used to represent the output perturbation).
  - Tables 2 and 3 lack citation in the main body of text, while Table 4, situated in the supplementary material, is referenced. This continually disrupts the reading rhythm.

---

> ### Author Response · Authors · 2024-01-12
>
> Dear Reviewer dDd7,
>
> Thank you for your time in reviewing our paper. We will start by addressing your main concern:
>
> *why the ability of accurately storing data enhances the performance in tasks such as the wind nowcasting.*
>
> Our study primarily focuses on tasks involving extrapolation, such as wind prediction and heat equation modeling using the Poisson model. These tasks are about extrapolation – we have a context, and we seek to extrapolate beyond the points where we have data. Extrapolation can be seen as a form of generalized interpolation. In particular, we want our model to have the same properties as a general interpolation scheme. Specifically, the model should match or be able to match the given interpolated values at the points where we have measurements. The rationale behind our set of experiments is that a model incapable of interpolating correctly will likely face problems when extrapolating.
>
> We do not believe that maintaining disentangled latent representations necessarily leads to more memorization. In fact, the model does not need to memorize the context data, as it has access to it at the time of prediction. What we demonstrate here is that some architectural choices might prevent the model from effectively using this data. Our experiments show that baseline models, in some scenarios, fail to generate an output function matching the provided context points. As you mentioned, this could act as a form of regularization, where their underfitting inadvertently helps to avoid overfitting. However, we argue that this is an undesirable property in the context of our tasks.
>
> Regarding your comment on multimodal models, some models, like [1,2], indeed fuse multimodal inputs to handle multimodal signals. However, these fusions are carefully designed so that inputs corresponding to the same events are combined and processed together, or to merge the outputs of two modal-specific models into a general one. In our case, the issue arises when the fusion of contexts occurs with inputs that should not be mixed, leading to a loss of information.
>
> Our experimental setup, especially in training and perturbation correction, demonstrates that this property can slow down the training process. We believe that the superior performance we observe is partly attributed to this aspect. Our ablation studies further emphasize our model's ability to interpolate and extrapolate effectively, showcasing its capacity for more efficient error correction during training.
>
> We have corrected Section 5.2. In figure6, GEN 5x5 and 6x6 actually took the same number of gradient updates (48), compared to only 39 for MSA. We have also addressed the presentation issues, and we deeply appreciate your pointing them out. We will send the updated version once we receive the last review, in accordance with the TMRL recommendation.
>
> [1] Jaegle, A., Gimeno, F., Brock, A., Vinyals, O., Zisserman, A., & Carreira, J. (2021, July). Perceiver: General perception with iterative attention. In International conference on machine learning (pp. 4651-4664). PMLR.
>
> [2] Fayek, H. M., & Kumar, A. (2020). Large scale audiovisual learning of sounds with weakly labeled data. arXiv preprint arXiv:2006.01595.

---

> ### Author Response · Authors · 2024-01-16
>
> Dear reviewer dDd7, now that the discussion period has started, we sent an updated version and highlighted the changes corresponding to your remarks. We did change the notation following your advice, we added definition of the missing terms and used $\epsilon$ instead of $\gamma$ in section 5.2.

---

### Review · Reviewer_8Rzh · 2024-01-13

**Summary Of Contributions:**

The work proposes a new method for modeling an unstructured set of measurements sparsely placed in space and time. The proposed architecture is transformer-based and addresses the challenge of bottlenecks in the learned representation in existing methods for such data.

The performance of the proposed method is validated on four problem domains: high-altitude wind nowcasting, two-day weather forecasting, fluid dynamics, and heat diffusion.

The results show improved performance against two existing methods; Graph Element Networks and Conditional Neural Processes. Moreover, the authors included in the analysis a transformer-based method with a representation bottleneck as an additional baseline to demonstrate the advantage of avoiding the representational bottleneck.

In addition to the analysis of real-world datasets and simulated dynamical systems, the authors present an additional experiment where the model is used to do context retrieval. The goal of this experiment is to show that sufficiently capable models can perfectly retrieve data from context while models that have a representational bottleneck struggle to achieve the same performance when the data has high spatial frequencies.

Finally, the authors present an analysis of the capability of the different architectures to do error correction when exposed to localized perturbations.

**Audience:**

Yes

**Broader Impact Concerns:**

No concerns about ethical implications.

**Claims And Evidence:**

No

**Requested Changes:**

- Please address or respond to all the indicated weaknesses given above. Particularly the perturbation analysis and Figure 5.
- Overall the current version of the paper does not meet the criteria for "clearly written" and as such I will answer the question on "Claims And Evidence" with a "No" at this time.

**Strengths And Weaknesses:**

**Strengths:**
- The approach is well-motivated
- The empirical analysis is done on a broad range of problem domains including publically available datasets
- The analyzed baseline models sufficiently demonstrated the advantages of the proposed approach
- Additional analysis is proposed for context retrieval and more saliently illustrates the limitations that arise from bottlenecks in the representation of models for sparse data.

**Weaknesses:**
- The presentation is somewhat messy. Specifically, the notation introduced in (1)-(4) operates per measurement, then in (5) $c_e$ is over the whole set. In (5) $c_l$ does not seem to have a role.
- The explanation of GEN (10)-(13) seems superficial and insufficient to understand the method without referring to the original publication. Such a half-detailed presentation does not play a good role in the paper.
- For CNP equation (15) $t_l$ is undefined.
- In the experiments section GKA method is mentioned. It is unclear why. It is not a baseline in the paper.
- The figures are poorly distributed throughout the paper. The reader needs to search for the figures and find them far from where they are referenced.
- Table 2. What does size refer to? The number of parameters of the model?
- The captions of the figures and the axis sometimes miss relevant information. E.g. Figure 3 is about wind forecasting, this is not indicated, nor is the metric in the vertical axis.
- Table 4 is referred to in the main body, but the table is actually in the appendix.
- What is the metric used in Figure 4?
- Why introduce the GNG and the PER model in the 5.1? What is the role of these models in this analysis?
- The term disentangled representation is used in this case when referring to a representation that this distributed over space. This term is already used in the field of "Learning disentangled representation" which has a different meaning. I think this can be misleading and a more precise term should be used.
- I fail to understand Figure 5 and Section 5.2. It is not clear to me how the perturbation is done and the error is backpropagated. I do not understand what the implication of this analysis is.
- There are several grammatical errors throughout the paper.

---

> ### Author Response · Authors · 2024-01-16
>
> Dear Reviewer 8Rzh,
>
> We sincerely apologize for any lack of clarity in our initial presentation. In response to your feedback, we have thoroughly revised the PDF document, highlighting the changes made. We hope these updates address your concerns effectively, and we are open to any further feedback should there still be areas of confusion.
>
> In response to the specific issues you identified, we have made the following adjustments in our revised document:
>
> - We have standardized the notation by consistently using sets, and we have enhanced the clarity regarding point (5).
> - The GEN baseline has been rewritten to include comprehensive details for a better understanding of the model.
> - The CNP notation has been completed.
> - For consistency, we compared our model to GKA, as it was the reference model in the wind nowcasting dataset study. However, GKA is not applicable to other experiments due to differences in input and output space dimensions.
> - We have relocated the figures to positions closer to where they are mentioned in the text.
> - The term 'size' corresponds indeed to the number of parameters, and this change has been reflected in the main table.
> - Additional details have been incorporated into the relevant captions.
> - We introduced a summarizing table in Section 5.3 to negate the need for referring to the comprehensive results table in the appendix.
> - The metric used is MSE and we added this information to the caption.
> - We introduced GNG and PER to highlight that the failure in the task was attributable solely to the bottleneck in the architecture, with expanded details provided in Section 5.1.
> - We appreciate your comment on the use of the term "disentangled latent representation". We reviewed the literature, and we agree that the term can be misleading. To improve clarity, we have replaced it with "independent".
> - We agree that section 5.2 could have been more clear. We want to see the effect of the bottleneck of the baselines on the training dynamics, and to do so we conduct an ablation restricting to a simple case where all the models have to learn to correct a single error in their outputs. We did a major rewrite of this section.
> - We tried to find the mistakes you mentioned and to correct them
>
> We thank you for your valuable input and hope that our revisions meet your expectations. Please do not hesitate to reach out if you have any further questions or require additional clarifications.

---

### Decision · Action_Editor_DYij · 2024-03-13

**Recommendation:** Accept as is

**Comment:**

Problem statement is interesting and relevant. Both physics based and machine learning baselines are evaluated. The model is simple but effective.

**Audience:**

This article is likely to be of interest to a broad audience within the TMLR community.

**Claims And Evidence:**

The work introduces a novel method for learning from sparse and unstructured data. The proposed method surpasses the physical and machine-learned baselines evaluated in the paper. The authors have improved the presentation in light of the reviews. At the end of the rebuttal period, all of the reviewers recommended publication.